



# Impacts of water partitioning and polarity of organic compounds on secondary organic aerosol over Eastern China

Jingyi Li[1, 2], Haowen Zhang[2], Qi Ying[3,*], Zhijun Wu[4, 1], Yanli Zhang[5,6], Xinming Wang[5,6,7], Xinghua Li[8], Yele Sun[9], Min Hu[4, 1], Yuanhang Zhang[4, 1], Jianlin Hu[1, 2, *]

[1] Collaborative Innovation Center of Atmospheric Environment and Equipment Technology, Nanjing University of Information Science & Technology, Nanjing 210044, China

[2] Jiangsu Key Laboratory of Atmospheric Environment Monitoring and Pollution Control, School of Environmental Science and Engineering, Nanjing University of Information Science & Technology, Nanjing 210044, China

[3] Texas A&M University, College Station, Texas 77843, USA

[4] State Key Joint Laboratory of Environmental Simulation and Pollution Control, College of Environmental Sciences and Engineering, Peking University, Beijing 100871, China

[5] State Key Laboratory of Organic Geochemistry and Guangdong Key Laboratory of Environmental Protection and Resources Utilization, Guangzhou Institute of Geochemistry, Chinese Academy of Sciences, Guangzhou 510640, China

[6] Center for Excellence in Regional Atmospheric Environment, Institute of Urban Environment, Chinese Academy of Sciences, Xiamen 361021, China

[7] University of Chinese Academy of Sciences, Beijing 100049, China

[8] School of Space & Environment, Beihang University, Beijing 100191, China

[9] State Key Laboratory of Atmospheric Boundary Layer Physics and Atmospheric Chemistry, Institute of Atmospheric Physics, Chinese Academy of Sciences, Beijing 100029, China

Corresponding authors:
Qi Ying, Email: qying@civil.tamu.edu
Jianlin Hu, Email: jianlinhu@nuist.edu.cn



**Abstract**
Secondary organic aerosol (SOA) is an important component of fine particular matter
(PM$_{2.5}$) in China. Most air quality models use an equilibrium partitioning method along
with estimated saturation vapor pressure of semi-volatile organic compounds (SVOCs) to
predict SOA formation. However, this method ignores partitioning of water vapor to the
organic aerosols and the organic phase non-ideality, both of which affect the partitioning
of SVOCs. In this study, the Community Multi-scale Air Quality model (CMAQv5.0.1)
was used to investigate the above impacts on SOA formation during winter (January) and
summer (July) of 2013 over eastern China. The organic aerosol module was updated by
incorporating water partitioning into the organic particulate matter (OPM) and considering
non-ideality of organic-water mixture. The modified model can generally capture the
observed organic carbon (OC), the total organic aerosol (OA) and diurnal variation of PM$_{2.5}$
at ground sites. SOA concentration shows significant seasonal and spatial variations, with
high concentration levels in North China Plain (NCP), Central China and Sichuan basin
(SCB) areas during winter (up to 25 μg m$^{-3}$) and in Yangtze River Delta (YRD) during
summer (up to 12 μg m$^{-3}$). When water partitioning is included in winter, SOA
concentrations increase slightly, with the monthly-averaged daily maximum relative
difference of 10-20% at the surface and 10-30% for the whole column, mostly due to the
increase in anthropogenic SOA. The increase in SOA is more significant in summer, by
20-90% at the surface and 30-70% for the whole column. The increase of SOA over the
land is mostly due to biogenic SOA while the increase of SOA over the coastal regions is
related with that of anthropogenic origin. Further analysis of two representative cities, Jinan
and Nanjing, shows that changes of SOA are favored under hot and humid conditions. The
increases in SOA cause a 12% elevation in the aerosol optical depth (AOD) and 15%
enhancement in the cooling effects of aerosol radiative forcing (ARF) over YRD in summer.
The aerosol liquid water content associated with OPM (ALW$_{org}$) at the surface is relatively
high over the land in winter and over the ocean in summer, with the monthly-averaged





daily maximum of 2-9 and 5-12 μg m$^{-3}$, respectively. By using the $\kappa$-Köhler theory, we
calculated the hygroscopicity of OA with modeled ALW$_{org}$, finding that the correlation
with O:C ratio varies significantly across different cities and seasons. Water partitioning
into OPM only promotes SOA formation, while non-ideality of organic-water mixture only
leads to decreases in SOA in most regions of eastern China. Water partitioning into OPM
should be considered in air quality models in simulating SOA, especially in hot and humid
environments.

**Keywords**: SOA, non-ideality, water partitioning, hygroscopicity

**1 Introduction**
Secondary organic aerosol (SOA) is formed via a complex interaction of volatile organic
compounds (VOCs) with oxidants and primary particles emitted from anthropogenic and
biogenic sources in the atmosphere. As an important component of fine particular matter
(PM$_{2.5}$), SOA can cause severe air pollution in urban and suburban areas (Huang et al.,
2014) and exhibit adverse health effects (Polichetti et al., 2009;Feng et al., 2016;Xing et
al., 2016;Atkinson et al., 2014). SOA also plays an important role in new particle formation
and particle growth (Man et al., 2015;Zhang et al., 2011;Wiedensohler et al., 2009;Yue et
al., 2011;Liu et al., 2014;Ehn et al., 2014;Huang et al., 2019;Jokinen et al., 2015) and
further contributes to the enhancement of cloud condensation nuclei (CCN) (Yue et al.,
2011;Wiedensohler et al., 2009;Liu et al., 2014;Jokinen et al., 2015). This will, in turn,
impact the atmospheric aerosol burden, precipitation and water circulation, solar radiation
budget, and climate (Rosenfeld et al., 2008;Spracklen et al., 2011;Quaas et al.,
2008;Ramanathan et al., 2001;Hatzianastassiou et al., 2007;Hegerl et al., 2015). However,
the mechanisms of these influences are not well understood so far, due to the high
uncertainties associated with the formation and physical and chemical properties of SOA
(Shrivastava et al., 2017). Large gaps still exist in SOA mass loadings and properties





between model estimates and laboratory and field measurements (Gentner et al.,
2017;Ervens et al., 2011;Hayes et al., 2015). Therefore, it is crucial to explore and resolve
this issue to improve our knowledge of the roles of SOA in the environment, health, and
climate.

Gas-particle partitioning of semi-volatile and low-volatile organic compounds

generated from VOC oxidation is an important pathway of SOA formation. In most current
chemical transport models (CTMs), this process is treated as an equilibrium partitioning
that depends on the mass concentration of organic particulate matter (OPM), ambient
temperature, the mean molecular weight of OPM, and the volatility of purer condensed
organics (Pankow, 1994). The volatilities of condensed organic products from a certain
precursor VOC are either represented by that of several lumped surrogates based on
chamber experiments (2-product model) (Odum et al., 1996) or fitted into different bins of
a fixed volatility range (usually $0.01\text{-}10^5$ µg m$^{-3}$) (volatility basis set model, VBS model)
(Donahue et al., 2006). Although the above models can capture the general trend of SOA
evolution and mass concentration to some extent (Slowik et al., 2010;Li et al., 2017a;Baek
et al., 2011;Bergström et al., 2012;Woody et al., 2016;Heald et al., 2006), both of them
neglected two key factors that may lead to biases: 1) the structures and interactions of
condensed organics (non-ideality); 2) partitioning of water vapor, an abundant atmospheric
constituent to OPM. The non-ideality alters the volatility of condensed organics, and thus
their contributions to the total SOA mass loading (Cappa et al., 2008). Water partitioning
into OPM can reduce the partial pressure of organics and lead to increase in SOA mass,
which is called the Raoult's Law effect (Prisle et al., 2010). This impact may vary for
different SOA precursors (Healy et al., 2009;Prisle et al., 2010). The above two aspects
will not only affect the chemical composition of SOA but also the inorganic portion (Ansari
and Pandis, 2000;Meyer et al., 2009) and optical properties (Liu and Wang, 2010;Denjean
et al., 2015) of aerosols.





Laboratory and field studies have confirmed the fact that water absorbed by SOA
(quantified as hygroscopicity, κ) from a variety of VOCs (Lambe et al., 2011;Zhao et al.,
2016b;Asa-Awuku et al., 2010;Varutbangkul et al., 2006). The hygroscopicity of SOA is
highly correlated with the oxygen-to-carbon ratio (O:C) and increases with more oxidized
SOA during photochemical aging (Poulain et al., 2010;Wang et al., 2014;Lambe et al.,
2011;Tritscher et al., 2011a;Zhao et al., 2016b;Massoli et al., 2010;Tritscher et al.,
2011b;Duplissy et al., 2011). The OPM-associated water partitioning can be estimated
using the κ-Köhler theory under the Zdanovskii-Stokes-Robinson (ZSR) assumption of no
interactions between any constitutes in aerosols (Petters and Kreidenweis, 2007). The total
water content is the summarization of each constitute at the same RH. Guo et al. (2015)
found that this simplified method, along with the ISORROPIA model which is used to
predict aerosol liquid water (ALW) associated with the inorganic portion of aerosols,
reproduced the observed total ALW in the ambient environment. Pye et al. (2017) applied
this approach along with a parameterization of overall $\kappa$ based on O:C ratio and a
simplified method to estimate activity coefficients of organics and found that modeled OA
and ALW are improved during daytime but still biased low at nighttime. Shortcomings still
exist in the above method for water associated organics (ALW$_{org}$) as interactions between
organic species in the organic-water mixture are not considered, which has been shown to
play an important role in SOA formation and water partitioning to OPM (Kim et al., 2019).
A representation of water partitioning along with SVOCs with consideration of water-
organic and organic-organic interactions in CTMs showed significant influences in SOA
and ALW in the eastern U.S. where biogenic SOA dominated in OA and the internal mixing
assumed for the aerosol (Pankow et al., 2015;Jathar et al., 2016).
China has been suffering from severe PM$_{2.5}$ pollution especially in the eastern region
with fast urbanization and economic development (Guo et al., 2014;Fu and Chen,
2017;Yang et al., 2016). The secondary portion has been proved to be dominated in PM$_{2.5}$
and organic aerosol increases during haze events (Huang et al., 2014;Sun et al., 2019). In





addition, SOA is a very important component of PM$_{2.5}$ in China that contributes about 20-
50% (Li et al., 2017b). Previous modeling studies indicated that SOA was underpredicted
in this region (Wang et al., 2011;Lin et al., 2016;Jiang et al., 2012) and the impacts of non-
ideality and water-OPM partitioning have not been evaluated.
In this study, regional simulations of SOA during January and July of 2013 over
eastern China under several scenarios were conducted to investigate the seasonal variation
of SOA due to water partitioning into OPM. Model performances were firstly evaluated
against observed meteorological parameters (temperature and relative humidity) as well as
PM$_{2.5}$, OC, and OA at ground monitoring sites. Then, the regional and seasonal impacts on
SOA and water content were quantified. Factors related to the impacts on SOA, including
sources of precursors, chemical compositions and meteorological conditions were further
analyzed. Lastly, the impacts on aerosol optical properties and hygroscopicity were
investigated.
**2 Methodology**
The Community Multi-scale Air Quality model (CMAQ v5.0.1) coupled with a modified
SAPRC-11 was used in this study. Model configurations were largely based on that used
by Hu et al. (2016) as summarized below. Firstly, SAPRC-11 was expanded for more
detailed treatment of isoprene oxidation and tracking dicarbonyl (glyoxal and
methylglyoxal) products from different groups of major precursors (Ying et al., 2015);
Secondly, heterogeneous formation of secondary nitrate and sulfate from NO$_2$ and SO$_2$
reaction on particle surface (Ying et al., 2014), and SOA from isoprene epoxydiols
(IEPOX), methacrylic acid epoxide (MAE) and dicarbonyls through surface-controlled
reactive uptake (Li et al., 2015;Pankow et al., 2015) were added; Thirdly, SOA yields were
corrected for vapor wall loss (Zhang et al., 2014).
Two types of SOA were considered in the current model, "semi-volatile" (SV) portion
that formed via equilibrium absorption-partitioning of SVOCs, and "non-volatile" (NV)
portion that formed via direct oxidation of aromatics at low-NO$_x$, isoprene oxidation

under





acidic conditions, reactive uptake of dicarbonyls, IEPOX and MAE, and oligomers. The
SV-SOA module mostly based on that of Pankow et al. (2015) with several updates in the
treatment of primary organic aerosol (POA) by including it in the non-ideality calculation
of the organic-water mixture. The mass distribution of SVOCs between the gas-phase and
particle-phase follows the equation:

$$K_{p,i} = \frac{F_i}{M \cdot A_i} \qquad (Eq\ 1)$$

where $K_{p,i}(m^3 \mu g^{-1})$ is the gas/particle partitioning constant for compound i, $F_i(\mu g\ m^{-3})$
is the concentration of species i in the particle phase, $A_i(\mu g\ m^{-3})$ is the concentration of
species i in the gas phase, and $M(\mu g\ m^{-3})$ is the total mass concentration of the absorbing
phase. The gas/particle partitioning constant $K_{p,i}$ is dependent on the composition of the
absorbing organic phase. Pankow et al. (1994) derived $K_{p,i}$ for SVOCs partitioning into
an absorbing organic phase as:

$$K_{p,i} = \frac{RT}{10^6 \overline{MW} \xi_i p_{L,i}^o} \qquad (Eq\ 2)$$

where $p_{L,i}^o(atm)$ is the saturation vapor pressure of the pure compound i at temperature
T(K), $\xi_i$ is the activity coefficient of species i in the absorbing organic phase, $\overline{MW}$ (g mol$^{-1}$
$^1$) is the average molecular weight of the absorbing organic phase, R (8.314 J mol$^{-1}$ K$^{-1}$) is
the gas constant, and $10^6$ is used to convert the unit to $m^3 \mu g^{-1}$.
There are 12 lumped SVOCs generated by oxidation of alkanes, alkenes, and
aromatics oxidized under different NO$_x$ conditions (Table S1). Activity coefficients of
SVOCs were calculated based on the composition of absorbing organic phase using the
UNIversal Functional Activity Coefficient (UNIFAC) method (Fredenslund et al., 1975),
with assigned carbon number (n$_c$), functional groups and energy interaction parameters to
both SV and NV compounds (Pankow et al., 2015). The UNIFAC model is one of the
commonly used models that activity coefficients of condensed organics and their
interactions with water can be estimated. This method has been adopted to investigate the
impacts of non-ideality and water-OPM partitioning on SOA for different precursors in





box models (Seinfeld et al., 2001;Bowman and Melton, 2004) and CTMs (Jathar et al.,
2016;Pankow et al., 2015;Kim et al., 2019). The primary organic aerosols (POA) was
assumed to have a bulk composition of ten categories of surrogate species (Table S3), as
used by Li et al. (2015). POA is also involved in the calculation of activity coefficients for
the organics in the condensed phase. Detailed information about the surrogate species
including the structures and properties can be found in Li et al. (2015) and references
therein.

In addition to organic compounds, water partitioning into OPM is enabled according

to Eq 1 and Eq 2. In such a case, the absorbing phase in Eq 1 includes both organic aerosols
and water partitioning into OPM. As water considered in the absorbing phase, it will further
alter the molar fraction of each composition, the activity coefficient of SVOCs and the SV-
SOA mass concentrations as a result.

As the water partitioning into OPM is highly correlated with the hygroscopicity of

aerosols ($\kappa$), their correlation can be expressed by the $\kappa$-Köhler theory with Kelvin effect
neglected (Peter et al., 2006):

$$ALW_{org} = V_{org}\kappa_{org}\frac{a_w}{1-a_w} \tag{Eq3}$$

where $V_{org}$ is the volume concentration of organic, and $a_w$ is the water activity (assumed to
be the same as RH). Taken the density of organic aerosol to be 1.2 g cm$^{-3}$ (Li et al., 2019),
the hygroscopicity of the total OA can estimated. This simplified method can be used to
estimate OPM associated water (Guo et al., 2015;Li et al., 2019). In addition, the
hygroscopicity of organic aerosol is dependent on the degree of oxygenation, showing a
positive linear relationship with the O:C ratio (Massoli et al., 2010;Duplissy et al.,
2011;Lambe et al., 2011;Hong et al., 2018;Li et al., 2019). We therefore estimated the
correlation of $\kappa$ and O:C ratio at 9 representative cities during January and July with the
reduced major axis regression method (Ayers, 2001). O:C ratio of the total OA was
calculated as following:




$$O:C = \sum_{i=1}^{n} f_i (O:C)_i \qquad (\text{Eq4})$$

where $f_i$ and $(O:C)_i$ are the molar fraction and O:C ratio of organic aerosol component
i. For POA, a fixed molar fraction and composition has been assumed following Li et al.
(2015). For SOA, the O:C ratio was estimated by their OM:OC ratio (Simon and Bhave,

2012):

$$O:C = \frac{12}{15}(OM:OC) - \frac{14}{15} \qquad (\text{Eq5})$$

OM:OC ratio of each SOA component follows Pye et al. (2017).
The simulation domain has a horizontal resolution 36 km × 36 km and a vertical
structure of 18 layers up to 21 km, which covers eastern China as shown in Figure S1.
Anthropogenic emissions were generated from the Multi-resolution Emission Inventory for
China (MEIC) (Zhang et al., 2009;Li et al., 2014;Zheng et al., 2014;Liu et al., 2015) v1.0
with a 0.25°× 0.25° resolution (http://www.meicmodel.org) for China, and the Regional
Emission inventory in Asia version 2 (REAS2) (Kurokawa et al., 2013) with a 0.25°× 0.25°
resolution (http://www.nies.go.jp/REAS/) for the rest of the domain. Biogenic emissions
were generated by the Model for Emissions of Gases and Aerosols from Nature (MEGAN)
v2.1, with the leaf area index (LAI) from the 8- day Moderate Resolution Imaging
Spectroradiometer (MODIS) LAI product (MOD15A2) and the plant function types (PFTs)
from the Global Community Land Model (CLM 3.0). Open biomass burning emissions
were generated from the Fire INvnetory from NCAR (FINN) (Wiedinmyer et al., 2011).
Dust and sea salt emissions were generated in line during CMAQ simulations.
Meteorological fields were generated using the Weather Research and Forecasting (WRF)
model v3.6.1 with initial and boundary conditions from the NCEP FNL Operational Model
Global Tropospheric Analyses dataset. More details about the model application can be
found in Hu et al. (2016)
Four scenarios are investigated in this study. The base case (BS) that applied the
default secondary organic aerosol module of CMAQ; the water case (S1) that only water





partitioning into OPM was considered; the UNIFAC case (S2) that effects of molecular
structure of the primary and secondary organic species were included; and the combined
case (S3) that S1 and S2 were combined together.
**3 Results**
**3.1 Model evaluation**
Temperature and relative humidity (RH) are the two meteorological factors that affect SOA
formation. Table 1 shows the comparison of WRF predictions and observations in 8 sub-
regions of the domain (Figure S1). Observed data are accessible from the National Climatic
Data Center at ftp://ftp.ncdc.noaa.gov/pub/data/noaa. Temperature and RH are well
captured by WRF in YRD, the Pearl River Delta (PRD), and central regions of China (the
major regions of eastern China). Model estimates of daily organic carbon (OC) from the
BS case were compared with measurements at monitoring sites in Beijing and Guangzhou
during the winter of 2013 (Figure 1(a)). Overall, the ratio between modeled and observed
OC concentration falls in the range of 1:2 to 2:1, with a correlation coefficient R of 0.70.
The model tends to underestimate OC, especially in Beijing on highly polluted days (by -
37~48%). No significant improvements to modeled OC were observed in S3. The impacts
of water co-condensation and polarity of organic condensed species on SOA exhibit strong
seasonal and spatial features, which are further discussed in Section 4. The impacts in
Beijing and Guangzhou are not significant during winter. The bias in OC might be due to
under-estimated POA emissions and under-predicted SOA in CMAQ from missing
precursors (Hu et al., 2017;Zhao et al., 2016a).
The model estimate of OA was further investigated. As shown in Figure 1(b), CMAQ
can well capture the observed diurnal variation of OA at Beijing during wintertime, except
for the underestimates of peak values. A better agreement between the model and the
observations is observed on non-polluted days (daily-averaged concentration less than 75
$\mu g\ m^{-3}$). The monthly-averaged mean fractional bias (MFB) and mean fractional error
(MFE) are -0.13 and 0.27, respectively. POA is the primary contributor to OA at Beijing





in winter, accounting for 88% due to aging of POA not treated in the current model. The
fraction of SOA is small, resulting in little impacts on SOA by water partitioning into OPM
and insignificant improvements of the modeled OA in S3.

Figure S2 shows the comparison of modeled and observed PM$_{2.5}$ at monitoring sites

as shown in Figure S1 (a) during July of 2013. Generally, our model can well reproduce
the diurnal variation of PM$_{2.5}$ in most regions. Predicted PM$_{2.5}$ on high concentration days
are biased low compared to observations, especially in the North Central Plain (NCP). The
NCP region has the highest PM$_{2.5}$ from 60 µg m$^{-3}$ to 300 µg m$^{-3}$ compared to other regions.
The bias in modeled PM$_{2.5}$ is significant in cities in the Northwest. This might be due to
missing dust emissions in the current inventory (Hu et al., 2016). To further evaluate the
model performance, statistics of MFB and MFE were plotted against observed PM$_{2.5}$
concentration at all monitoring sites (Figure S3). The criteria and goal followed
recommendations of Boylan and Russell (2006). Our model performed well as most of the
predictions meet the criteria and a large fraction (>58%) meet the goal. The averaged MFB
and MFE are -0.28 and 0.39 respectively, indicating slightly underestimate of PM$_{2.5}$ by the
model.
**3.2 Impacts of water partitioning on SOA**
Distribution of SOA varied greatly in the two seasons. In winter, SOA is relatively high in
eastern SCB and in the contiguous areas of Shandong, Henan, Anhui, and Hubei provinces
(Figure 2 and Figure S4). Monthly-averaged SOA concentrations in the above two areas
are up to 25 and 15-20 µg m$^{-3}$, respectively. The major precursors of SOA are originated
from anthropogenic sources such as dicarbonyl products of aromatics oxidation, xylenes
and toluene (Figure S5). In summer, surface SOA is high in NE, NCP and YRD regions.
Shanghai, Jiangsu province and coastal areas of Yellow Sea show the highest SOA of ~9-
12 µg m$^{-3}$ at the surface and ~20 mg m$^{-2}$ as the column total (col-SOA) in the atmosphere
below 21 km (Figure S4). Different from winter SOA, a significant fraction of summer



SOA is originated from biogenic emissions in Shanghai and Jiangsu province (Figure S5).
Anthropogenic SOA is high in July in coastal areas of Yellow Sea and Bohai Bay.

Combined effects of water partitioning into OPM and non-ideality on SOA formation

(S3) also exhibit strong seasonal variation. In winter, the increase of SOA is relatively
small, by ~1-4 μg m$^{-3}$ (10-20%) at the surface (Figure 2) and less than ~5 mg m$^{-2}$ (10-30%)
as for the column concentration (Figure S4). The influences on SOA also differ in different
altitudes. For example, the maximum increment at the surface is observed in Shandong
province in NCP (Figure 2), while SOA at higher levels of the atmosphere is more
significant in South China (Figure S4). The increase in SOA is mostly attributed to
anthropogenic sources in winter (Figure S5 and S7). In summer, higher temperature and
relative humidity (RH) promote SOA formation as well as water partitioning into OPM. At
the surface, SOA increases by 3-9 μg m$^{-3}$ (40-50%) in coastal areas and 2-9 μg m$^{-3}$ (20-
90%) over the land, which are dominated by anthropogenic and biogenic origin,
respectively (Figure S6). For col-SOA, in addition to coastal areas, more significant
increase is observed in YRD, most of Henan province, and the contiguous areas of Hubei,
Hunan, and Jiangxi province (Figure S4) by about 30-70%.

Regional distribution of water partitioning into OPM is similar to the changes of SOA.

Figure 3 shows the regional distribution of monthly-averaged daily maximum ALW$_{org}$. We
see up to 9 μg m$^{-3}$ ALW$_{org}$ at surface occurs in Shandong in winter where great increment
in SOA appears as well. In other areas, ALW$_{org}$ is about 2-6 μg m$^{-3}$. The ratio of ALW$_{org}$
to SOA is about 0.1-0.5 in winter. In summer, water partitioning mostly involves in east
coastal areas at the surface where significant increase of anthropogenic SOA (such as
toluene and xylenes) is observed. This might be due to the high polarity of anthropogenic
SVOCs (having more -COOH groups) that absorb more water. In the coastal areas, ALW$_{org}$
is about 5-12 μg m$^{-3}$, with a ratio to SOA of 0.3-0.6. ALW$_{org}$ over the land is about 2-7 μg
m$^{-3}$ (ALW$_{org}$/SOA ratio of 0.1-0.4) in most areas, which is mostly associated with the
increase of BSOA such as isoprene and monoterpenes with abundant OH group in SVOCs.





The highest ALW$_{org}$ is 16 μg m$^{-3}$ near Shanghai (ALW$_{org}$/SOA ratio of 0.57). Water
partitioning also varies at different altitudes (Figure S9). In winter, more column water
partitions into OPM (col-ALW$_{org}$) in Chongqing, Hunan, Guanxi, Guangdong and Guizhou
province, with the col-ALWorg/col-SOA ratio of 0.2-0.3. In summer, higher col-ALW$_{org}$
is predicted over the land, especially in YRD, with the col-ALW$_{org}$/col-SOA ratio of 0.1-
0.3 over eastern China.
Figure 4 shows the correlation of $\kappa_{org}$ with O:C ratio. The estimated O:C ratio is
within the range of 0.2-0.6. In summer, the oxidation state of OA shows different degrees
of enhancement compared to winter at most of the cities except Guangzhou, due to
increased contribution of SOA to total OA. The averaged $\kappa_{org}$ of OA in each O:C bin
falls in the range of 0.001-0.1, with the highest $\kappa_{org}$ (~0.3) at Beijing in summer. The
linear correlation between $\kappa_{org}$ and O:C shows significant spatial and seasonal variations.
For example, the slope of $\kappa_{org}$-O:C is much smaller in winter (45-74% less) than in
summer in the Northern cities such as Shenyang, Beijing, Zhengzhou, and Xi'an, while the
slope of $\kappa_{org}$-O:C in winter is much higher (47-104% more) than in summer in the
Southern cities, such as Nanjing, Chengdu and Guangzhou. In Jinan and Shanghai, the
slope is quite similar in both seasons. The fitted correlations are very different from
previous studies with a relatively higher slope of $\kappa_{org}$-O:C from 0.18 to 0.37 (Duplissy et
al., 2011;Lambe et al., 2011;Massoli et al., 2010;Chang et al., 2010), indicating the
hygroscopicity of aerosols with chemical complexity cannot be simply represented by a
single parameter such as O:C (Rickards et al., 2013).
**3.3 Impacts on solar radiation**
The impacts on aerosol optical depth (AOD) and aerosol radiative forcing (ARF) were
further investigated. Figure 5 shows the monthly-averaged AOD at 550 nm in January and
July of 2013. It was calculated as the accumulation of model estimated extinction
coefficient of fine particles (*EXT$_i$*) multiplied by the thickness (*HL$_i$*) of each layer:


$$AOD = \sum_{i=1}^{N} EXT_i \times HL_i \qquad \text{(Eq6)}$$

Where N is the number of layers. There are two methods to estimate the aerosol extinction
coefficient in CMAQ. One is using the Mie theory ($EXT_m$), and the other is based on
extinction values from the IMPROVE monitoring network that considers the impacts of
hygroscopicity of different aerosol components ($EXT_r$)(Malm et al., 1994). AOD calculated
with the two types of extinction coefficient are denoted as $AOD_m$ and $AOD_r$, respectively.

In Figure 5, a clear pattern of high $AOD_r$ in SCB and NCP and low $AOD_r$ in west

China in both winter and summer is observed, consistent with previous studies (He et al.,
2019;He et al., 2016;Luo et al., 2014). An identified trend in $AOD_m$ is observed as shown
in Figure S10. The monthly-averaged $AOD_r$ ranges from 1.1 to 3.5 in January and from 0.4
to 0.8 in July. $AOD_m$ is lower than $AOD_r$, falling in 0.7-2.2 in January and 0.3-0.6 in July.
The model significantly overestimates AOD in January but agrees better with observations
from MODIS in the high regions in July (Figure S11). Biases in the predicted AOD might
be partially due to the empirical equation applied in the calculation of AOD in CMAQ
(Wang et al., 2009;Liu et al., 2010), and partially due to the uncertainties of fine AOD
overland from MODIS data (Wang et al., 2009;Levy et al., 2010). With water partitioning
into OPM, changes in SOA mass concentration and chemical composition lead to increase
of AOD, which shows a strong spatial and seasonal pattern. In winter, there is no significant
increase in $AOD_r$ across the whole domain, due to insignificant changes of SOA. In
summer, $AOD_r$ increases in YRD and the adjacent area of Hubei, Hunan, and Jiangxi
province by up to 12%.

ARF represents the changes in the radiative flux due to aerosols. The off-line version

of the Shortwave Radiative Transfer Model For GCMs (RRTMG_SW) is used to calculate
the direct radiative effect of aerosols on shortwave radiation (Iacono et al., 2008). Generally,
fine aerosols exhibit cooling effects on the shortwave radiation in both winter and summer
over the entire domain as shown in Figure 6. This impact is much stronger in the areas





where AOD is high (Figure 5). The ARF at top of atmosphere (TOA) is highest in
Shandong in winter and coastal areas near Jiangsu province, which are about -12 W m$^{-2}$
and -9 W m$^{-2}$, respectively. In winter, no significant changes of ARF are observed in the
high regions of eastern China (Figure 6). This is likely attributed to an insignificant
contribution of SOA to PM$_{2.5}$ in winter compared to other components with cooling effects,
such as sulfate. In summer, SOA is an important component of PM$_{2.5}$ (20-60%), and the
effects of water partitioning on shortwave radiation is relatively stronger. An enhancement
of up to 15% in the cooling effects of ARF occurs near YRD region where AOD
significantly changes as well.
**4 Discussion**
Meteorological conditions and SOA precursors affect the impacts of water partitioning on
SOA. Figure 7 shows the effects of different factors on the daily maximum change of SOA
in Jinan and Nanjing, two representative cities in winter and summer, respectively. As
shown in Figure 7(a), the daily maximum elevation of SOA occurs when RH is greater than
70% in both cities. This is consistent with the previous study in the Southeast U.S. during
summer (Pankow et al., 2015). A clear correlation of the changes in SOA with SOA
concentration in Nanjing (R=0.84) during summer can be observed. However, this
correlation is relatively weak in Jinan (R=0.44) during winter. There is no strong
correlation between changes in SOA and temperature as shown in Figure 7(b), likely due
to the daily variation of SOA mass and composition. To better illustrate the dependency of
SOA on temperature and relative humidity, an offline calculation of SOA formation was
performed at Jinan and Nanjing when the daily maximum SOA increases occurred. We
assumed temperature (T) and water vapor mixing ratio (QV) to be within the range of $\bar{X} \pm$
$\sigma$, where $\bar{X}$ and $\sigma$ are the mean and standard deviation calculated based on WRF
prediction at each location. We chose 10 evenly distributed values for T and QV within the
range of $\bar{X} \pm \sigma$. The temperature dependence parameter of saturation vapor pressure ($\Delta H$)
was also scaled by 0.2, 0,8, 1.4 and 2.0 separately for all the SVOCs. As shown in Figure



8, SOA indicates a negative correlation with temperature and a positive correlation with
RH in both cities. SOA is more sensitive to RH under cool conditions (JN) and to
temperature under hot conditions (NJ). An interesting finding is that significant increases
in SOA in the two cities occur during different time periods of the day. Water partitioning
tends to affect SOA in the afternoon and evening in Jinan, which mostly happens in the
early morning and at noon in Nanjing. The different timing is likely attributed to a
substantial increase in SOA precursors in the two cities. In Jinan, the most contributing
SVOCs are originated from toluene and xylenes oxidation, as well as oligomers formed by
their oxidation products in OPM. Possible emission sources include transportation,
petroleum refining, manufacturing, painting, etc. SOA increase in Nanjing is mostly
associated with biogenic sources including isoprene and monoterpenes.

Impacts of water partitioning into OPM and non-ideality of organic-water mixture on

SOA are opposite. Water partitioning alone increases SOA by ~20-60% in winter and ~20-
100% in summer (Figure S12). This is because that the molecular weight of water is quite
small and will reduce the molar averaged weight of OPM ($\overline{MW}$) in Eq 2 (Pankow et al.,
2015). The reduced $\overline{MW}$ further increases $K_{p,i}$ promoting the mass transfer of SVOCs
from the gas phase to the OPM. On the other hand, by considering non-ideality of organic-
water mixture, activity coefficients of SVOCs are usually greater than 1.0 in this study,
leading to a decrease in $K_{p,i}$. As a result, the total SOA concentration is reduced by up to
~10% in winter and ~30% in summer in the high regions (Figure S13). Overall, the final
impacts are the combined consequences of the two "processes". In winter, the increase of
SOA caused by water partitioning is offset by the decrease of SOA due to the polarity of
SVOCs in most areas of the domain, resulting in no significant changes. In summer, effects
of water partitioning overcome that of SVOC polarity so as the total SOA loading increases.
This further leads to an enhanced attenuation of shortwave solar radiation and cooling of
the atmosphere.
**5 Conclusion**





The WRF/CMAQ model was used to investigate the impacts of water partitioning into OPM and non-ideality of organic-water mixture on SOA formation over eastern China during January and July of 2013. SOA is greatly enhanced in summer especially in YRD and over Yellow Sea by up to 90% and 70% at the surface and the whole column, respectively. No significant impacts on SOA are observed in winter. ALW$_{org}$ is highly correlated with the changes of SOA, with the ratio of ALW$_{org}$ to SOA of 0.1-0.5 and 0.1-0.6 at the surface where significant changes of SOA occur in winter and summer, respectively. By using the modeled ALW$_{org}$, correlations between $\kappa_{org}$ and O:C were examined in 9 representative cities, showing significant spatial and seasonal variations. The increases in SOA lead to 12% elevation of AOD and 15% enhancement in the cooling effects of ARF in summer. The effects of water partitioning into OPM and non-ideality of organic-water mixture on SOA were also examined separately. Since the activity coefficients of SVOCs are mostly greater than 1.0 during the simulated episode, SOA concentrations decrease when non-ideality effect is considered. Daily SOA concentration decreases by up to ~10% in winter and ~30% in summer in the high regions. Water partitioning alone increases SOA by ~20-60% in winter and ~20-100% in summer. It should be noticed that the results shown in this study are the lower limit as the current model tends to underestimate SOA. It is crucial to consider both effects in simulating SOA formation under hot and humid conditions in CTMs.

*Data availability*. Data used in this manuscript can be provided upon request by e-mail to the corresponding authors Qi Ying (qying@civil.tamu.edu), and Jianlin Hu (jianlinhu@nuist.edu.cn).

*Author contributions*. J.L. and Q.Y. revised the model. J.L. performed the simulations. Yan.Z., X.W., X.L. and Y.S. provided observations of OC and OA. J.L. and H.Z. processed and analyzed the data. J.L., H.Z., Q.Y., J.H. and Z.W. discussed the results. J.L., Q.Y. and



J.H. contributed to writing and editing of the manuscript with comments from all the co-
authors.

*Competing interests*. The authors declare that they have no conflict of interest.

*Acknowledgments*. This project was partly supported by National Key R&D Program of
China (2018YFC0213802, Task #2), National Science Foundation of China (No. 41705102
and 41875149), and the Major Research Plan of National Social Science Foundation
(18ZDA052). The authors thank James F. Pankow for providing SOA module code. Jingyi
Li acknowledges support from the Startup Foundation for Introducing Talent of NUIST
grant no. 2243141701014, the Priority Academic Program Development of Jiangsu Higher
Education Institutions (PAPD), and Six Talent Peaks Project of Jiangsu Province.

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





**Table 1.** Statistical analysis of modeled temperature (K) and relative humidity (%) of January and July at the monitoring sites in different geographical areas as shown in Figure S1.

| | | Northeast | | NCP | | Northwest | | YRD | | Central | | Sichuan Basin | | PRD | | Southwest | |
|---|---|---|---|---|---|---|---|---|---|---|---|---|---|---|---|---|---|
| | | Jan | Jul | Jan | Jul | Jan | Jul | Jan | Jul | Jan | Jul | Jan | Jul | Jan | Jul | Jan | Jul |
| Temperature (K) | Obs | 256.7 | 296.4 | 264.4 | 297.9 | 268.1 | 294.0 | 277.8 | 303.6 | 276.1 | 301.8 | 276.9 | 296.6 | 290.3 | 301.1 | 282.1 | 294.6 |
| | Pred | 251.6 | 296.5 | 261.3 | 298.9 | 268.5 | 295.0 | 278.5 | 302.8 | 276.3 | 301.5 | 274.5 | 294.0 | 289.4 | 300.2 | 278.4 | 291.1 |
| | MB | -5.1 | -0.1 | -3.1 | 1.3 | 0.48 | 1.0 | 0.7 | -0.8 | 0.2 | -0.3 | -2.5 | -2.6 | -0.8 | -1.0 | -3.7 | -3.5 |
| | GE | 6.3 | 4.4 | 5.1 | 5.1 | 6.1 | 5.4 | 3.6 | 4.2 | 4.5 | 4.4 | 6.7 | 5.4 | 3.5 | 2.7 | 6.1 | 4.4 |
| | Num | 10101 | 11298 | 15072 | 16820 | 11475 | 12830 | 6835 | 7620 | 21210 | 23809 | 13573 | 15192 | 7017 | 7715 | 12088 | 13590 |
| Relative Humidity (%) | Obs | 77.9 | 80.4 | 74.9 | 72.4 | 61.0 | 70.8 | 79.3 | 69.5 | 76.9 | 71.2 | 69.5 | 74.2 | 76.0 | 80.8 | 69.6 | 78.7 |
| | Pred | 85.3 | 73.4 | 78.4 | 58.1 | 48.7 | 56.1 | 73.6 | 73.0 | 64.5 | 70.3 | 57.8 | 78.4 | 75.2 | 84.5 | 78.0 | 87.0 |
| | MB | 7.4 | -7.0 | 3.5 | -14.3 | -12.3 | -14.6 | -5.6 | 3.4 | -12.4 | -0.9 | -11.6 | 4.2 | -0.8 | 3.7 | 8.4 | 8.2 |
| | GE | 12.4 | 19.2 | 16.2 | 22.7 | 21.3 | 22.6 | 17.6 | 17.8 | 20.8 | 17.9 | 25.5 | 17.2 | 15.0 | 13.8 | 20.6 | 16.2 |
| | Num | 10101 | 11298 | 15072 | 16820 | 11475 | 12830 | 6835 | 7620 | 21210 | 23809 | 13573 | 15192 | 7017 | 7715 | 12088 | 13590 |

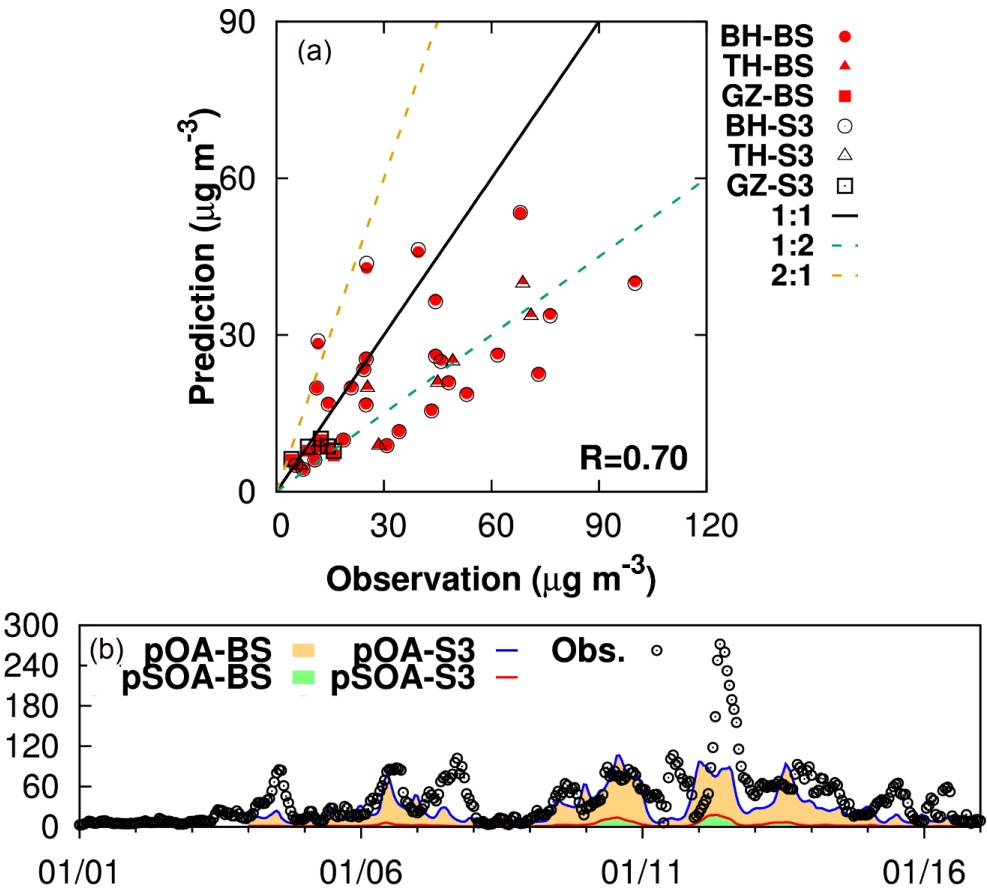

**Figure 1**. Comparison of (a) observed and modeled organic carbon concentration at University of Beihang (BH), Tsinghua University (TH) and Guangzhou (GZ); (b) observed organic aerosol (Obs.) at Beijing and predictions of total OA (pOA) and SOA (pSOA), unit is μg m$^{-3}$. Locations of monitoring sites are shown in Figure S1.

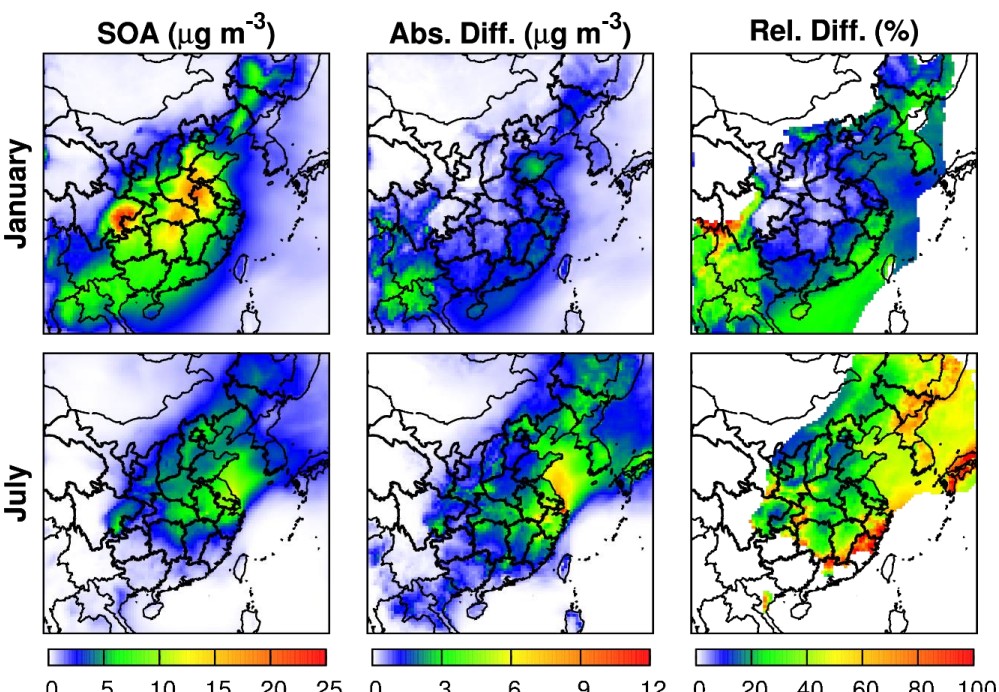

**Figure 2**. Monthly-averaged total SOA in BS and monthly-averaged daily maximum changes of SOA due to water partitioning and non-ideality of organic-water mixture. "Abs. Diff." represents absolute differences (S3-BS); "Rel. Diff." represents relative differences ((S3-BS)/BS, %). Relative differences are shown in areas with monthly-averaged SOA concentration greater than 1 µg m⁻³.

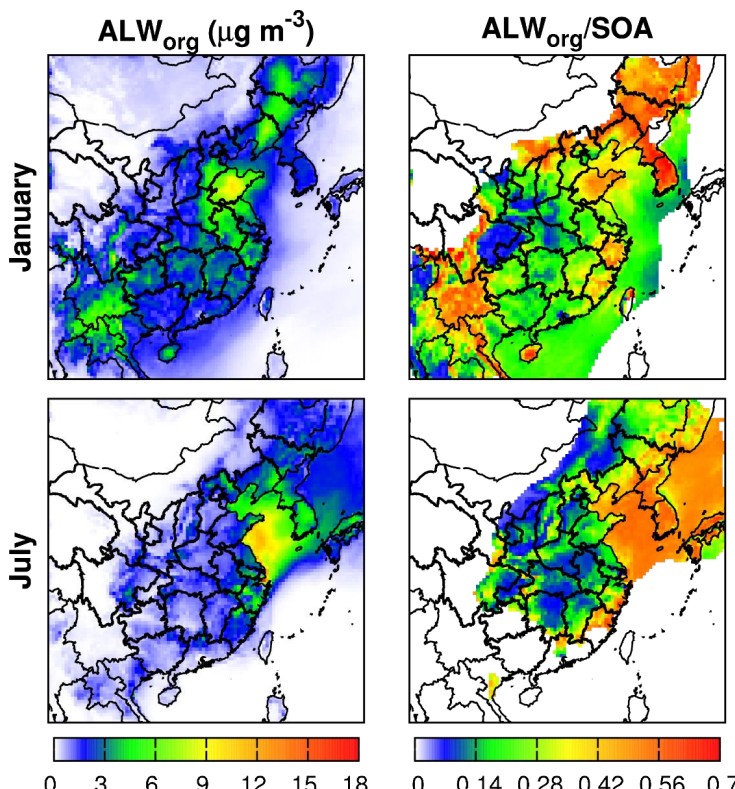

**Figure 3**. Monthly-averaged daily maximum water partitioning into the organic-phase (ALW$_{org}$, µg m$^{-3}$) and the ratio to SOA (ALW$_{org}$/SOA) during January and July of 2013. AWL$_{org}$/SOA is shown in areas with monthly-averaged SOA concentration greater than 1 µg m$^{-3}$.

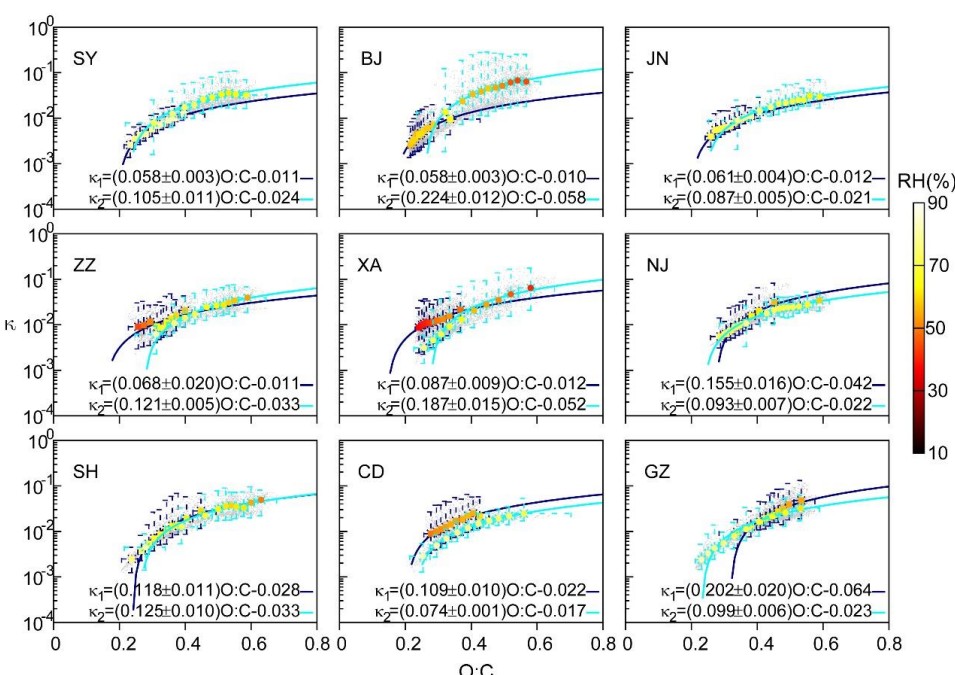

**Figure 4** The correlation of hygroscopicity of aerosol ($\kappa$) and O:C ratio at 9 representative cities including Shenyang (SS), Beijing (BJ), Jinan (JN), Zhengzhou (ZZ), Xi'an (XA), Nanjing (NJ), Shanghai (SH), Chengdu (CD), and Guangzhou (GZ). Gray dots on the background represent all the data in January and July, which are categorized into several O:C bins. In each bin, the ranges of $\kappa$ and O:C ratio are represented by dashed bars colored for January (navy) and July (light blue), with the mean value colored by the averaged RH of each bin. The mean $\kappa$ and O:C ratio are fitted by reduced major axis regression.

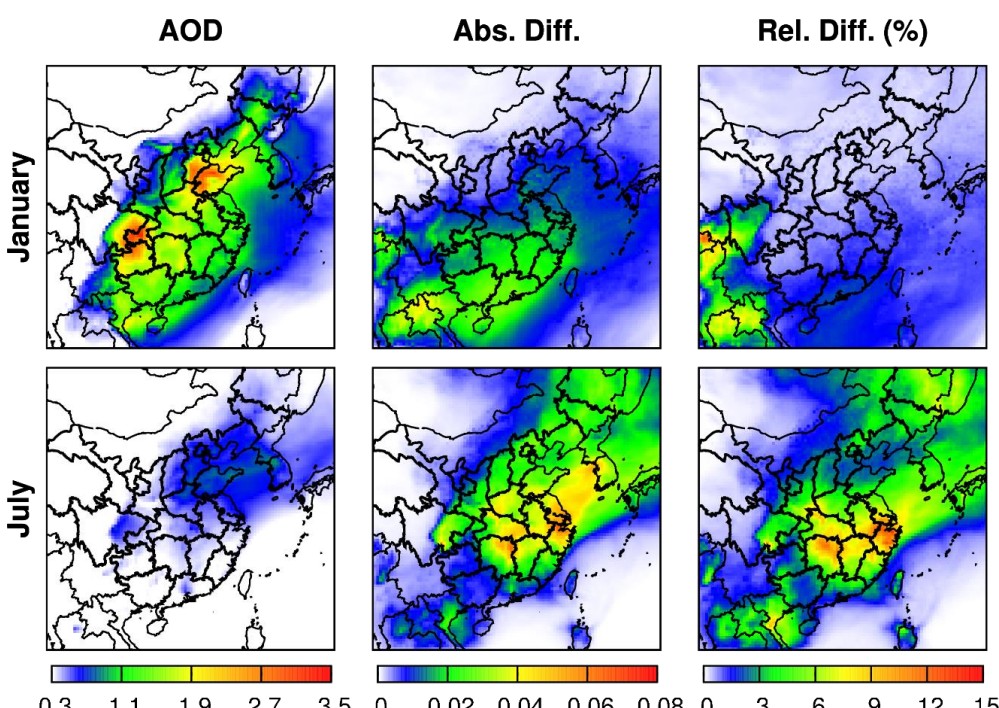

**Figure 5**. Monthly-averaged $AOD_r$ at 550 nm and the monthly-averaged daily maximum changes of $AOD_r$ due to water partitioning and the non-ideality of organic-water mixture. "Abs. Diff." represents absolute differences (S3-BS); "Rel. Diff." represents relative differences ((S3-BS)/BS, %).

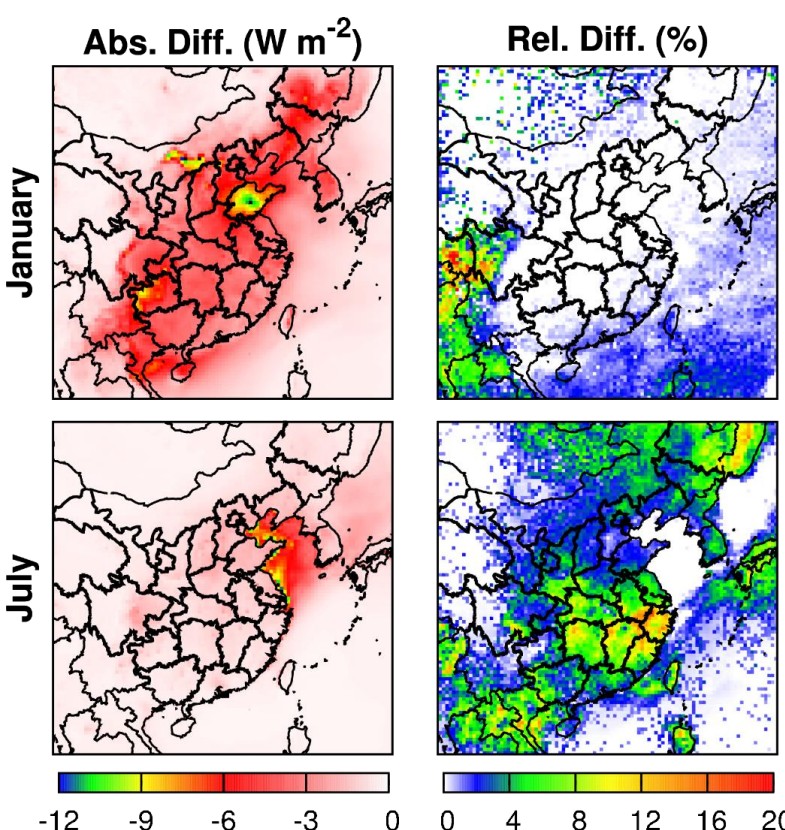

**Figure 6.** Monthly-averaged daily maximum variation of shortwave direct aerosol radiative forcing at the top of atmosphere due to water partitioning during January and July of 2013. "Abs. Diff." represents absolute differences (S3-BS); "Rel. Diff." represents relative differences ((S3-BS)/BS, %).

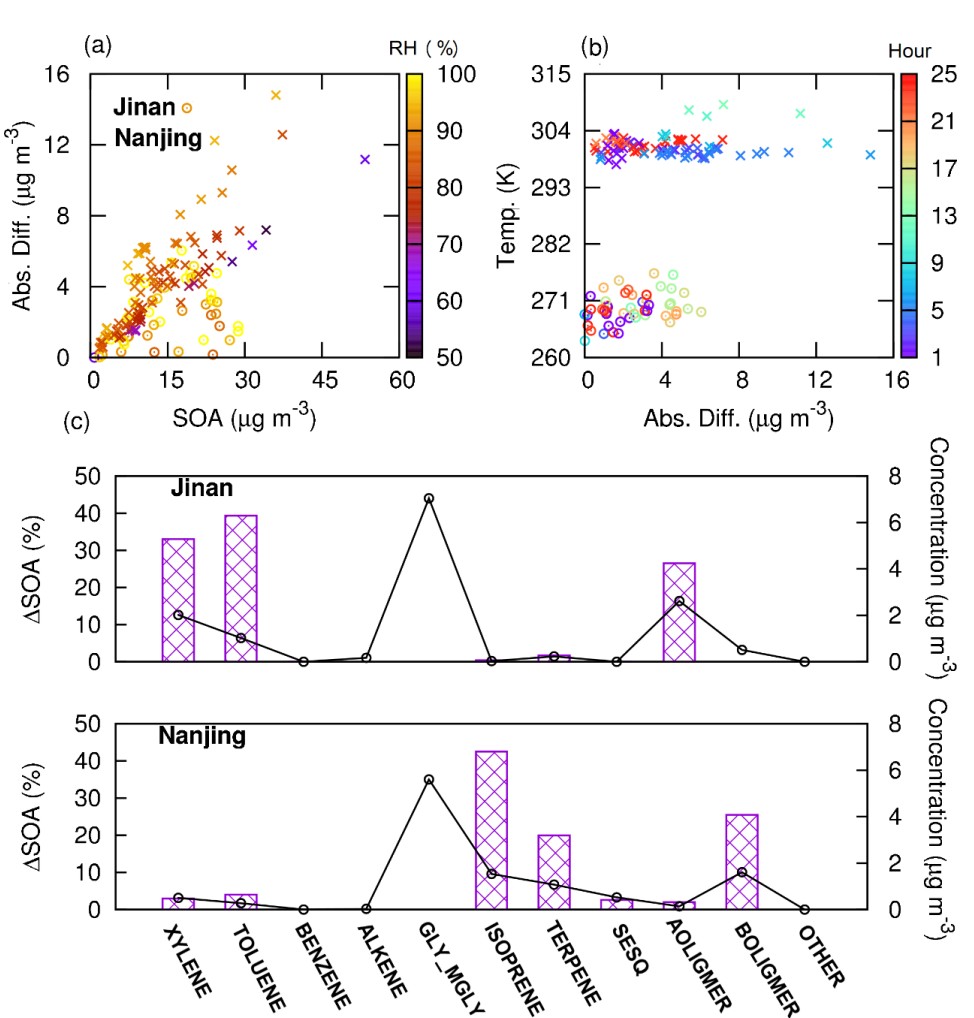

**Figure 7**. Correlation of water partitioning on SOA with (a) RH (b) temperature at Jinan in winter and Nanjing in summer, and the contribution from each SOA component to the total SOA increase. In plot (a) and (b), "Abs. Diff." represents the daily maximum change of SOA that is calculated as S3-BS. Color box represents RH in (a) and the hour in the day in (b) when daily maximum change of SOA occurred. In (c), the left axis represents contribution of each SOA component to the daily maximum SOA change due to water partitioning, and the right axis represents the concentration of each SOA component.





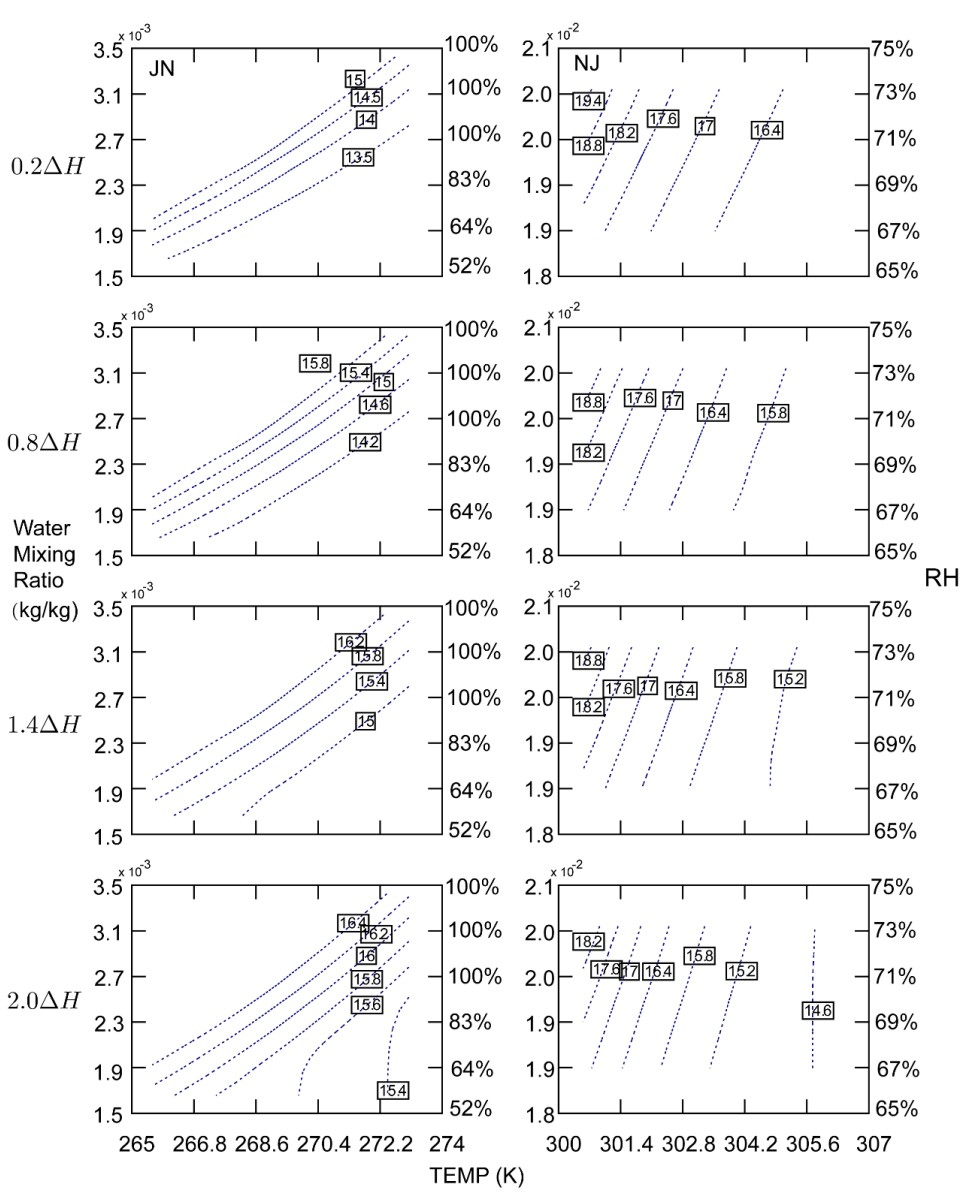

**Figure 8.** Sensitivity of SOA formation to temperature (TEMP), relative humidity (RH) and the temperature dependence parameter of SVP ($\Delta H$) at Jinan (JN, first column) and Nanjing (NJ, second column). The relative humidity is showing on the right side of y-axis.