# Peer review of "Impacts of water partitioning and polarity of organic compounds on"

_Atmospheric Chemistry and Physics, 2019_

## Referee Comment (RC1) · Anonymous Referee #1 · 25 Feb 2020

The authors present model results using the WRF/CMAQ for SOA formation in China. The model is updated for the partitioning of water vapor to the organic aerosol and the non-ideality of the organic phase. A comparison of the model to observations is performed for multiple sites. SOA enhancement during the summer and winter is discussed for the different China domains. The effect of aerosol liquid water on SOA formation and aerosol optical depth are presented. Correlation of the calculated OA hygroscopicity based on the k-Köhler theory to the O:C ratio is performed to show seasonal and multicity variations. Although the publication could provide valuable insights into the factors that govern SOA formation in China it currently lacks a detailed discussion and validation of the presented results. Therefore, the publication requires major

revisions suggested below.

Major Comments

The manuscript is hard to read. Discussion and Results are not well separated in the manuscript. Many phrases are not clear and require more elaboration and better use of English. The table and most figures are poorly made and the science is hard to follow. Examples are given below.

A major drawback in this work is that the model is not capturing the SOA formation during the winter that has been shown to be the dominant organic aerosol source in multiple publications for different domains of China. The authors only in a sentence discuss that the conversion of the POA to SOA may be the reason for these discrepancies but they have no observations to back this up.

There is no discussion on the influence of nitrate on aerosol liquid water. What fraction of the ALW is related to nitrate and what to organics? How could the ALW related to nitrate influence the partitioning of organics?

A comparison of the model to observations should be performed and presented in the main text for both seasons in more detail. The effect of the improvements performed for the SAPRC-11 model is not discussed. The processes added e.g., the heterogeneous formation of nitrate and sulfate on the particle surface, SOA from isoprene, and dicarbonyls surface-controlled reactive uptake are not discussed. What is the effect of these added processes to the overall performance of the model? Each addition and the effect should be discussed in detail in order to support the importance of including them.

MEGAN has been shown to overestimate the isoprene emissions. Would this have a major effect on SOA formation in this work?

Table 1: There is no discussion of the table in the main text. Abbreviations are not included in the caption. What is MB, GE, Num? Discussion on more statistically relevant values would be beneficial, e.g. what is the R2 of the comparisons? What are the presented values? Medians? Means? What is the domain of the model and how much do the values fluctuate around the domain? What are the uncertainties of the measurements and the model? A figure of the comparison of temperature and relative humidity of obs. vs model with the associated errors and linear regression analysis with the statistics would be informative.

Figure 1: The current Figure has no information regarding the season the measurements are performed. A comparison of both seasons should be made and a figure like Fig1(a) should be made for each season. The timeseries should include the same site for the two seasons. Why is the base case exactly the same as the S3 in Figure 1(b)? Add errors to the measurements.

Figure 2: Why do you use in (a) the base case and not the S3 case? I would consider promoting the updated S3 on the left and the changes on the middle and right panels.

Figure 3: I don't see the point in presenting the ratio of LWCorg to SOA. If both are expected to increase during pollution episodes then the ratio might stay the same therefore providing no valuable information. I would plot the SOA to OA as an alternative option or the SOA alone.

Figure 4: The data are really hard to observe. Please change colors and increase font size.

Figure 7: The graph is not clear. In the main text, the authors discuss that the daily maximum of SOA occurs when RH is greater than 70% in both cities. The RH is higher than 70% all the time. The time of the day is up to 24 hours and not 25. The markers and boxes for (c) are not discussed whether they represent the left or right axis.

Specific Comments

Line 34: Please elaborate more

Line 39: Please define generally with a statistical value that has meaning.

Line 91: Please elaborate more on the "purer condensed organics" for non-experts.

Line 99: "neglect" instead of "neglected".

Line 99: Please elaborate more on 1).

Line 109: The sentence is missing a verb.

Line 128-131: The sentence is hard to read. Please rephrase.

Line 138: Which region?

Line 163: Don't change a line. Also, why acidic conditions? Please, elaborate more. Why is the reactive uptake of dicarbonyls, IEPOX and MAE in the "non-volatile" category?

Line 164: change to "was mostly"

Line 165: Please elaborate more on the non-ideality calculation of the organic-water mixture for non-experts.

Line 170: Is this the absorbing organic phase?

Line 196: Change to "as water condenses".

Line 200: Please elaborate more on the "Kelvin effect neglected" for non-experts.

Line 204: Change to "can be estimated".

Line 242: Observations in 8 sub-regions of the domain during which period?

Line 251: No significant improvements observed when applying the above additions means that the model is still missing a significant pathway to SOA formation, especially since OA in both seasons are dominated by SOA based on observations. This should be discussed in detail and in the context of previous studies and findings from AMS measurements in China.

Line 258: Does it capture the observed diurnal variation? What is the R2 or R of

the timeseries of the modeled to observed values? What is the ratio of the two? In many cases, it seems that the difference is higher than a factor of 4. Is that a usual discrepancy? If so, how is much is it improved when incorporating the detailed SOA models?

Line 259: It would be great if "better" was described with statistical terminology. A way to describe the data and the comparison to modeled values would be to generate box and whiskers of the ratio of observations to modeled values for non-polluted and polluted days, respectively.

Line 262-263: POA is not the primary contributor to OA in Beijing in winter. Many studies show that SOA is the major contributor and the path towards SOA formation is currently unknown and strongly dependent on LWC in the particles. Aging of POA not treated in the model is not guaranteed to be the main source of SOA.

Line 280: Here only one season is provided in terms of timeseries comparison of the model and obs. Please provide both seasons.

Line 285: Figure S5 shows the anthropogenic SOA and not the dicarbonyl SOA. Please separate the contributions and discuss them in the main text. Identifying the contribution of different compounds to SOA formation in China would be of great interest to the scientific community.

Line 312: What about particulate nitrate?

Line 380: RH is higher than 70% all the time. What is the meaning of this sentence?

---

## Referee Comment (RC2) · Anonymous Referee #2 · 13 Mar 2020

In this paper, Li et al. have modified the CMAQ model to take into account the impacts of water partitioning and polarity of organic compounds on SOA formation. The model was applied over Eastern China to estimate the regional and seasonal impacts of these modifications on SOA and the aerosol water content. This study may have potential to contribute in the organic aerosol modeling field but major revisions needs to be done before publication. In particular, I have several concerns regarding the validity of the scientific methodology used and the presentation of the study. Therefore, I would recommend publication only if these comments will be addressed and fundamental changes will be contacted.

Major comments

Page 4 line 94: The majority of current CTMs have replaced the 2-product model with the VBS approach. Please make this clear and refer to the 2-product model of Odum et al. (1996) for historical reasons.

Page 6, line 162: Please add the appropriate references to support the nonvolatile nature of the products by these oxidation pathways.

Page 6 line 149: There is no discussion in the methodology about the observations and the statistical analysis metrics used to evaluate the model performance. Especially for the OA observations, there is no reference provided or description of the methods used.

Page 7 line 179: More details are needed here. How the model defines the low NOx and high NOx conditions? Which compounds each of the lump species represent? What is the difference between the lumped species of the same precursor (e.g., BNZ1, BNZ2, BNZ3)? Can you include the aerosol yields for each lumped species in tables S1 and S2? Are these aerosol yields NOx-dependent? What does the SVP stands for in Tables S1 and S2?

Page 8, lines 194-198: Does the absorbing phase of equation 1 includes only the water associated with the organics (from eq. 3) or it includes the total water (including the water associated with the inorganic aerosol components)? Under high RH (higher than the organic/inorganic phase separation RH, SRH), the aerosol organic phase is well mixed with the inorganic salts and, therefore, the aerosol water associated with the inorganic constituents can also contribute to the SOA absorbing medium (Pye et al., 2017). Please clarify what you have assumed here and add the relative discussion.

Page 8, Equation 3: This equation gives the volume of water associated with the organic fraction of the aerosol. However, ALW on the left hand side refers to the mass of water. Please correct.

Page 8, lines 203-204: How do you estimate the hygroscopicity?

Page 8, lines 204-205: How do you calculate the ALW? Are you using $\kappa$org and the eq3? Do you use the kappa hygroscopicity to calculate the ALW? If so, how you estimate the kappa?

Page 9, lines 212-216: Add a reference to tables S1 and S2. Furthermore, why the values of OM:OC in tables S1/S2 are different than the values provided by your reference (Pye et al., 2017)?

Page 9 lines 217-218: Can you add the total size of the model domain?

Page 9 lines 233: What boundary conditions are used? Please make a comment on how these can affect the simulation results. I would recommend adding spatial maps of primary organic aerosol emissions and SOA precursor emissions and summarizing in a Table the domain average emission rates of POA and each SOA precursors.

Pages 9-10 lines 234-238: This paragraph needs to be expanded and written in a separate section. In this section the authors should describe in more detail the following: i) Basecase simulation. Please explain how the default CMAQ is simulating POA and SOA and how different is this modelling configuration with the one the authors are testing, ii) Sensitivity simulations. Please explain in much more detail the sensitivity simulations conducted in this work. In addition the authors say that they have conducted three sensitivity scenarios named S1, S2 and S3. In their manuscript they only show results from S3 and they never discuss the results of S1 and S2.

Page 10 line 240: The section "Model evaluation" is extremely problematic and raises questions on the validity of the modelling results given that the model evaluation is insufficient. More specifically: 1) Given that the CMAQ default configuration has been modified to consider the importance of water and organic compound polarity on SOA formation, an accurate evaluation of the updated model performance is needed. 2) You should compare the model results for organic compounds during both July and January 2013. Currently, the evaluation includes a comparison with OC observations only during January over only three locations of the relatively large model domain. Furthermore, please mention in the text what factor have you used to convert the modeled OA to OC. 3) The total PM2.5 measurements have been used to evaluate the model performance during July without explaining the rationale of this choice since the focus of this study is solely the organic fraction of the aerosols. I suggest removing the PM2.5 evaluation or at least moving it to the supplement. 4) Can you include more OC/OA observations over other areas of their domain in your evaluation? 5) It is also important to compare the simulated POA and SOA against observations (e.g., from AMS). Furthermore, it would be helpful to show how the model performance against SOA measurements changes between the BC and the S1, S2, S3 cases.

Page 10 lines 251-253: The authors state here that the impacts of water-co-condensation and polarity of organic condensed species on SOA formation are not significant during winter. This highlights the need to evaluate their model results during July where they have found significant changes with the basecase simulation. Furthermore, the results from the three sensitivity simulations should be evaluated individually.

Page 10 lines 254-256: These are indeed possible factors. Can the authors comment, based on their analysis, which of these two possible factors is more important and try to be more specific? A comparison with AMS observations would be helpful here.

Page 11 line 263: The aging of POA, under specific conditions can enhance the SOA formation, especially over polluted areas. Can the authors comment how this important omission of their model configuration can affect their result? Once again, a comparison against POA and SOA from AMS observations will be helpful to identify the limitations of their model due to the treatment of POA as non-volatile and non-reactive.

Page 11 line 264: Please provide two spatial maps of the fraction SOA/Total OA during January and July 2013 so as to show the contribution of POA and SOA to total OA during each simulation period.

Page 13 line 317: You need a zonal map to show how the water partition changes with altitude and not the total column.

Page 13, lines 322-336: It is not clear how you calculate the $\kappa$org in your model. This is very important for this section.

Figure 1: The quality of the figure is poor. It is extremely difficult to see all the plotting data and the changes due to the use of different scenarios (especially in figure 1b).

Figure 2: I found the use of daily maximum concentration in the "difference" maps misleading. Given that you have the monthly average SOA from the basecase simulation, I would prefer to see the absolute (and relative) change of the monthly average SOA due to the use of S3 as well, and not the daily maximum. Furthermore, please add (a), (b), etc. to each subplot of the figure and add this information to the figure caption (apply this change in the rest of the figures as well).

Figure 4: All the fitted correlations listed here suggest that compounds with very low or zero O:C have negative hygroscopicity. Can you comment on this limitation and include a discussion in the text?

Table S3. Please explain in the table what the fraction in the last column stands for. How have you estimated the molecular weight and fraction of the POA from unknown compounds?

Minor Comments

The language and structure of the sentences can be substantially improved in many parts of the manuscript. Just a few examples are listed here, but I suggest revising thoroughly the wording in the whole text.

Page 5, lines109-11: Please rephrase. The sentence sounds wrong.

Page 5, lines117-118: Please rephrase.

Page 6 line 144: OC and OA abbreviations have not been used before in the main text.

Page 8, line 204: Please correct the "can estimated" to "can be estimated".

Page 8 line 206: I would use the word "correlate" instead of "dependent"

Page 9 line 229: Change "in" with "on"

Page 10 line 254: Change the sentence to: "In Beijing and Guangzhou, these impacts are not significant during winter"

Page 11 line 274-275: Please rephrase.

Page 11 line 282: Which two areas? You have mentioned several areas in the previous sentence.

Page 13 line 317: "column water". Please rephrase

---

## Author Comment (AC1) · 16 Apr 2020

**Response to Referee #1**

The authors thank the reviewer for the comments that improve the quality of the paper. The detailed responses are given as follows. The reviewer comments are shown in italic font, the responses are in regular font, and the revised text is in bold font.

*The authors present model results using the WRF/CMAQ for SOA formation in China. The model is updated for the partitioning of water vapor to the organic aerosol and the non-ideality of the organic phase. A comparison of the model to observations is performed for multiple sites. SOA enhancement during the summer and winter is discussed for the different China domains. The effect of aerosol liquid water on SOA formation and aerosol optical depth are presented. Correlation of the calculated OA hygroscopicity based on the k-Köhler theory to the O:C ratio is performed to show seasonal and multicity variations. Although the publication could provide valuable insights into the factors that govern SOA formation in China it currently lacks a detailed discussion and validation of the presented results. Therefore, the publication requires major revisions suggested below.*

**Major Comments**

**Comment 1:** *The manuscript is hard to read. Discussion and Results are not well separated in the manuscript. Many phrases are not clear and require more elaboration and better use of English. The table and most figures are poorly made and the science is hard to follow. Examples are given below.*
**Response 1:** We thank the reviewer for pointing this out. We have updated all the figures and tables by showing monthly-averaged results instead of the daily maximum average as used in the previous version. Also, the text has been revised carefully to make it clear and easy to follow.

**Comment 2:** *A major drawback in this work is that the model is not capturing the SOA formation during the winter that has been shown to be the dominant organic aerosol source in multiple publications for different domains of China. The authors only in a sentence discuss that the conversion of the POA to SOA may be the reason for these discrepancies but they have no observations to back this up.*
**Response 2:** We agree with the reviewer that there is no direct observational evidence for this particular episode that POA aging played a significant role in the SOA formation. However, this process is one of the important missing

sources of SOA in several models, field and chamber studies (Robinson et al., 2007; Shrivastava et al., 2008; Zhao et al., 2016; Jimenez et al., 2009; Hodzic et al., 2010; Murphy et al., 2017). Also, organic compounds with intermediate volatility (IVOCs) between SVOCs and traditional VOCs, especially from combustion sources might contribute to SOA as well (Robinson et al., 2007; Shrivastava et al., 2008; Tkacik et al., 2012; Zhao et al., 2014; Zhao et al., 2016; Hodzic et al., 2010), but with high uncertainties in the emission inventory and SOA yields (Shrivastava et al., 2008;  Tkacik et al., 2012). We performed an additional simulation using the latest version of the CMAQ model (v5.3.1), which includes a parameterization of these processes. The predicted SOA indeed increases significantly (Figure R1). We further analyzed the contribution of traditional POA (as semi-volatile POA in CMAQv5.3.1), SOA from the oxidation of the semi-volatile POA, traditional SOA, and a new SOA surrogate (pcSOA) representing missing SOA from IVOC oxidation, multigenerational aging of VOC oxidation products, and underestimate of SOA yield due to chamber wall losses (Murphy et al., 2017), finding pcSOA dominates in Beijing and Guangzhou (Figure R2) as well as the whole domain (Figure R1). The averaged SOA/POA ratio in Beijing is increased to 1.53, which is more consistent with field measurements (Zhao et al., 2019; Sun et al., 2016; Sun et al., 2013). However, the emission factors and oxidation rate of pcSOA precursors are highly uncertain and the contribution of pcSOA requires more observational constrains (Murphy et al., 2017).

We also examined the sensitivity of SOA and organic liquid water (ALW$_{org}$) to pcSOA and POA in an offline calculation in Beijing, Guangzhou, Jinan, and Nanjing. POA has the same properties as we used in the model. Non-volatile isoprene SOA is taken to represent pcSOA as their similarities in saturation vapor pressure and O:C ratio. We found that both SOA and ALW$_{org}$ are positively correlated with pcSOA, increased by 2-5 times in different locations when pcSOA increased by 2 times. The impacts of water partitioning into OPM and non-ideality of the organic-water mixture by including the above process should be explored in a future study.

We extend the discussion of potential reasons for underestimated SOA in the revised text (L309-326):
"**Again, no apparent changes of SOA nor OA are observed between case S3 and BS (not shown), since POA is predicted to be the primary contributor to OA at Beijing in winter in the current model, with an averaged SOA/POA ratio of 0.12. This ratio is much lower than the field observation of about 0.45-1.94 (Zhao et al., 2019; Sun et al., 2013; Sun et al., 2016). The bias might be due to the missing SOA converted by partitioning and aging of semi-volatile POA as well as oxidation from**

intermediate volatile organic compounds (IVOCs) and VOC oxidation products. Those pathways are shown to be important for SOA formation by modeling, field and chamber studies (Hodzic et al., 2010; Jimenez et al., 2009; Murphy et al., 2017; Robinson et al., 2007; Shrivastava et al., 2008; Tkacik et al., 2012; Zhao et al., 2014; Zhao et al., 2016a).

A sensitivity test was performed by using the newest CMAQ model version 5.3.1 that includes all the above processes in the aerosol module. The SOA/POA ratio in Beijing is improved greatly to be 1.53 in winter. However, high uncertainties still exist in the emissions of the involved precursors and characterization of SOA formation through these processes, needing further constrains by observations. Their influences on water partitioning into OPM and non-ideality of the organic-water mixture on SOA will be evaluated in a future study."

[Figure]

**Figure R1**. Mean SOA, SOA/OA, and pcSOA/SOA ratio predicted during January and July of 2013 by CMAQv5.3.1.

[Figure]

**Figure R2**. Modeled concentration of semi-volatile POA (sv-POA), OA and fraction of each organic aerosol component fsvPOA (sv-POA), foxPOA (oxidation of sv-POA), fpcSOA (pc-SOA) and fsvSOA (traditional SOA) in Beijing (a, c) and Guangzhou (b, d). Observations of OA in January 2013 at Beijing (Obs.) are also included in (a). The left axis is for the concentrations ($\mu$ g m$^{-3}$) and the right axis is the fraction of OA components.

**Comment 3:** *There is no discussion on the influence of nitrate on aerosol liquid water. What fraction of the ALW is related to nitrate and what to organics? How could the ALW related to nitrate influence the partitioning of organics?*

**Response 3:** In the current model, we separately treated the liquid water associated with organics (ALW$_{org}$) and inorganics (ALW$_{ing}$) in the condensed phase. Nitrate is assumed to only affect the inorganic aerosols and ALW$_{ing}$ as the interactions between inorganic and organic phases are not considered currently. This is the same approach used by Pankow et al. (2015). When considering the interactions between inorganic and organic aerosols in a CMAQ model, Pye et al. (2017) found an increase in SOA, which impacts are less significant than the separate treatment of the two phases. However, the interactions among condensed organics, i.e. the polarity of organics in the aerosol were ignored in their study. The interactions between inorganic and organic phases on ALW and SOA are beyond the scope of the current study and will be investigated in the future. We added a statement in the revised text (L207-L208) to make it clear:

"**In the current model, we assumed no interactions between the inorganic and organic phases.**"

**Comment 4:** *A comparison of the model to observations should be performed and presented in the main text for both seasons in more detail. The effect of the improvements performed for the SAPRC-11 model is not discussed. The processes added e.g., the heterogeneous formation of nitrate and sulfate on the particle surface, SOA from isoprene, and dicarbonyls surface-controlled reactive uptake are not discussed. What is the effect of these added processes to the overall performance of the model? Each addition and the effect should be discussed in detail in order to support the importance of including them.*

**Response 4:** The contribution of heterogeneous formation of nitrate and sulfate are not related to SOA formation. Papers documenting these changes have already been cited in the manuscript (Ying et al., 2014; Hu et al., 2016), and the impacts of the heterogeneous chemistry on nitrate and sulfate were discussed in those and another study (Zheng et al., 2015). The predicted nitrate and sulfate have been extensively compared with observations (Shi et al., 2017; Qiao et al., 2018). The improvements in the modeled SOA such as the reactive uptake of dicarbonyls and the isoprene generated epoxydiols have been discussed in previous studies by Ying et al. (2015), Li et al. (2015), Hu et al. (2017) and Liu et al. (2020) and have been shown to greatly increase the predicted SOA concentrations. The focus of this work is partitioning of water into OPM and the polarity of condensed organic compounds on SOA formation in China, which have not been examined so far. We have revised the manuscript to make it clear (L149-L161):

"**Model configurations were largely based on that used by Hu et al. (2016) as summarized below. Firstly, SAPRC-11 was expanded for a more detailed treatment of isoprene oxidation and tracking dicarbonyl products (glyoxal and methylglyoxal) from different groups of major precursors (Ying et al., 2015). Secondly, SOA from isoprene epoxydiols (IEPOX), methacrylic acid epoxide (MAE) and dicarbonyls through surface-controlled irreversible reactive uptake were added (Hu et al., 2017; Li et al., 2015; Liu et al., 2020; Ying et al., 2015). Thirdly, the heterogeneous formation of secondary nitrate and sulfate from NO2 and SO2 reaction on the particle surfaces (Ying et al., 2014) were added, which is an important source of secondary inorganic aerosols (Zheng et al., 2015) and improves model estimates of nitrate and sulfate (Qiao et al., 2018; Shi et al., 2017). Fourthly, SOA yields were corrected for vapor wall loss (Zhang et al., 2014). Impacts of the above updates on model performances have been extensively discussed in the cited work and will not be further investigated in the current study.**"

**Comment 5:** *MEGAN has been shown to overestimate the isoprene emissions. Would this have a major effect on SOA formation in this work?*

**Response 5:** The MEGAN model has been shown to overestimate emissions of isoprene in the eastern and southeastern US (Wang et al., 2017; Kota et al., 2015), which is mainly due to databases of emission factors used. While our previous study indeed showed up to 5 times higher isoprene concentrations compared to the observations in Nanjing, the isoprene oxidation products (MACR and MVK) agree well with observations. Another study of Southern China also showed up to 2 times higher of MEGAN compared to measured isoprene fluxes (Situ et al., 2014). Thus, it is still inconclusive whether isoprene was indeed overestimated. If this isoprene overestimation was prevalent throughout the country and the isoprene SOA changed linearly with isoprene emissions, the actual SOA concentration and impacts on SOA due to water partitioning will decrease by 40-50% and 20-30%, respectively. Emissions of isoprene and other biogenic emissions are low during winter so that little or no impact is expected for winter.

**Comment 6:** *Table 1: There is no discussion of the table in the main text. Abbreviations are not included in the caption. What is MB, GE, Num? Discussion on more statistically relevant values would be beneficial, e.g. what is the R2 of the comparisons? What are the presented values? Medians? Means? What is the domain of the model and how much do the values fluctuate around the domain? What are the uncertainties of the measurements and the model? A figure of the comparison of temperature and relative humidity of obs. vs model with the associated errors and linear regression analysis with the statistics would be informative.*

**Response 6:** Table 1 shows the mean observation (OBS), mean prediction (PRE), mean bias (MB), gross error (GE), and the number of valid data of temperature and relative humidity in 8 sub-regions of the domain as shown in Figure S1. The table has been revised with explanations of all the abbreviations. A new table (Table S5) has been added to explain each region in Table 1. Since there are too many observation sites in the domain to show the uncertainties and regression information, we added R in the revised Table 1 and updated the table. More information about measurement methodology and uncertainties associated with measurements can be found at the NCDC data website. We discussed Table 1 in the original text (L245-L249):

"Table 1 shows the comparison of WRF predictions and observations in 8 sub-regions of the domain (Figure S1). Observed data are accessible from the National Climatic Data Center at ftp://ftp.ncdc.noaa.gov/pub/data/noaa. Temperature and RH are well captured by WRF in YRD, the Pearl River Delta (PRD), and central regions of China (the major regions of eastern China)."

We expanded this discussion in the revised manuscript in L277-L288:

"**Temperature and RH are the two meteorological factors that affect SOA formation. Table 1 lists model statistics of mean observation (OBS), mean prediction (PRE), mean bias (MB), gross error (GE) and correlation coefficient (R) based on WRF and observations at monitoring sites located in 8 sub-regions of the domain (Figure S1) during January and July of 2013. The benchmarks for the MM5 model (another meteorology model) of 4-12km horizontal resolution suggested by Emery et al. (2001) are also listed in the table. Details of monitoring sites in the 8 sub-regions are listed in Table S5. Overall, WRF tends to underestimate both temperature and RH. The model shows better agreement with observed temperature as R is higher than that of RH. Both temperature and RH are well captured by the model in YRD, the Pearl River Delta (PRD), and the central regions of China (the major regions of eastern China). In these regions, MB and GE of temperature are -1.2~0.7 K and 1.8~2.6 K, respectively, which are -11.8~5.6% and 9.2~16.8% for RH, respectively.**"

**Comment 7:** *Figure 1: (1) The current Figure has no information regarding the season the measurements are performed. A comparison of both seasons should be made and a figure like Fig1(a) should be made for each season. The time series should include the same site for the two seasons. (2) Why is the base case exactly the same as the S3 in Figure 1(b)? (3) Add errors to the measurements.*

**Response 7:** (1) Due to limited observations of the simulated episode, we only have OC and OA measurements in January of 2013 at these sites. We used surface $PM_{2.5}$ alternatively to evaluate model performances in July of 2013. As a significant fraction of $PM_{2.5}$ in July is secondary, this still provides an indirect assessment of the model prediction of the oxidation capacity of the atmosphere, which is import for SOA formation. (2) The insignificant difference between BS and S3 in Figure 1(b) is likely due to a much smaller fraction of SOA compared to POA at this location predicted in the current model. Related discussions can refer to Response 2. (3) Unfortunately, there is no error information available for those measurements at this point.

**Comment 8:** *Figure 2: Why do you use in (a) the base case and not the S3 case? I would consider promoting the updated S3 on the left and the changes on the middle and right panels.*

**Response 8:** We have replaced case BS with S3 in Figure 2(a) and (d) and showed monthly-averaged results instead of averaged daily maximum in (b)-(c) and (e)-(f) in Figure 2.

**Comment 9:** *Figure 3: I don't see the point in presenting the ratio of LWCorg to SOA. If both are expected to increase during pollution episodes then the ratio might stay the same therefore providing no valuable information. I would plot the SOA to OA as an alternative option or the SOA alone.*

**Response 9:** One effect of water in the organic phase is that it decreases the average molecular weight of the absorbing organic phase, which could affect the subsequent partitioning of other semi-volatile organic compounds (see Eq 2). The ALW$_{org}$/SOA allows the readers to see the importance to consider ALW$_{org}$ in the partitioning calculation as it can account for a significant fraction of SOA and lead to a reduced average molecular weight. We agree with the reviewer that this ratio might not be very different between clean and polluted episodes. However, it is nonetheless useful in assessing the importance of including ALW$_{org}$ in SOA modeling. SOA/OA is certainly very useful information as well. Since POA is still significantly higher than SOA, especially during the winter month, SOA/OA ratio did not change significantly when ALW$_{org}$ is considered as shown in Figure R3. We have added this figure in the revised supplemental materials as Figure S13.

[Figure]

**Figure R3**. Averaged SOA/OA ratio from case BS and S3 during January and July of 2013.

**Comment 10:** *Figure 4: The data are really hard to observe. Please change*

*colors and increase font size.*

**Response 10:** Figure 4 (as shown below) has been revised to show all the data from each city in each month. Detailed results of each city are shown in Figure S9 and S10.

[Figure]

**Figure 4.** The correlation of hygroscopicity of organic aerosol ($\kappa_{org}$) and O:C ratio at 9 representative cities including Shenyang (SS), Beijing (BJ), Jinan (JN), Zhengzhou (ZZ), Xi'an (XA), Nanjing (NJ), Shanghai (SH), Chengdu (CD), and Guangzhou (GZ) in January (a) and July (b) of 2013. O:C ratios are categorized into 10 bins. In each bin, the ranges of O:C and $\kappa_{org}$ are represented by bars.

The mean values of O:C and $\kappa_{org}$ are represented by triangles colored by the averaged RH of each bin. The relationship between $\kappa_{org}$ and O:C is fitted by a linear function with reduced major axis regression (blue lines) and an exponential function (red lines), respectively. $\kappa_{01}$ and $\kappa_{07}$ represent the fitted correlation for January and July, respectively.

**Comment 11:** *Figure 7: The graph is not clear. In the main text, the authors discuss that the daily maximum of SOA occurs when RH is greater than 70% in both cities. The RH is higher than 70% all the time. The time of the day is up to 24 hours and not 25. The markers and boxes for (c) are not discussed whether they represent the left or right axis.*

**Response 11:** We have removed this figure to avoid confusion.

**Specific Comments**

**Comment 12:** *Line 34: Please elaborate more*

**Response 12:** Now the text reads:

"**However, the models typically assume that the organic particulate matter (OPM) is an ideal mixture and ignore the partitioning of water vapor to OPM.**"

**Comment 13:** *Line 39: Please define generally with a statistical value that has meaning.*

**Response 13:** The text has been revised in L40-42 as follows:

"**The modified model can generally capture the observed surface organic carbon (OC) with a correlation coefficient R of 0.7, and the surface OA with the mean fractional bias (MFB) and mean fractional error (MFE) of -0.28 and 0.54, respectively.**"

**Comment 14:** *Line 91: Please elaborate more on the "purer condensed organics" for non-experts.*

**Response 14:** This should be condensed organics and has been revised accordingly.

**Comment 15:** *Line 99: "neglect" instead of "neglected".*

**Response 15:** The text has been revised as instructed.

**Comment 16:** *Line 99: Please elaborate more on 1).*

**Response 16:** The text has been revised in L98-100 as following:

"*1) the molecular structures and interactions of functional groups (-OH, -C=O, -COOH, etc.) of condensed organics (non-ideality);*"

**Comment 17:** *Line 109: The sentence is missing a verb.*

**Response 17:** The sentence has been revised in L109-111 as following:

"**Laboratory and field studies have observed water absorbed by SOA from a variety of precursor VOCs (Lambe et al., 2011; Zhao et al., 2016b; Asa-Awuku et al., 2010; Varutbangkul et al., 2006).**"

**Comment 18:** *Line 128-131: The sentence is hard to read. Please rephrase.*

**Response 18:** The sentence has been revised in L126-129 as follows:

"*Using UNIversal Functional Activity Coefficient (UNIFAC) method (Fredenslund et al., 1975) for calculating activity coefficients of the organic-water mixture, it was found that in the eastern U.S., where biogenic SOA dominated the OA, considering ALW$_{org}$ leads to a*"

**significant increase in predicted SOA (Pankow et al., 2015; Jathar et al., 2016).***"*

**Comment 19:** *Line 138: Which region?*
**Response 19:** The region refers to China. The text has been revised in L134-135 as follows:
**"Previous modeling studies in China indicate that SOA was underpredicted (Lin et al., 2016; Jiang et al., 2012)"**

**Comment 20:** *Line 163: (1) Don't change a line. (2) Also, why acidic conditions? Please, elaborate more. (3) Why is the reactive uptake of dicarbonyls, IEPOX and MAE in the "non-volatile" category?*
**Response 20:** (1) This is a mistake due to file format conversion and has been corrected. (2) This is a typo. SOA formed by isoprene oxidation under acidic conditions refers to IEPOX and MAE SOA based on chamber experiments (Lal et al., 2012; Lin et al., 2012; 2013). We have removed "isoprene oxidation under acidic conditions" in the revised text. (3) In the current model, we assume that the reactive uptake of dicarbonyls, IEPOX and MAE is irreversible, as an upper-limit estimation of SOA from these precursors. That's why they are classified as non-volatile SOA. The text has been revised in L165-173 as following:
**"SOA from dicarbonyls, IEPOX, and MAE were formed by irreversible reactive uptake and categorized as NV-SOA in the current model as well. Some studies investigated SOA from glyoxal, methylglyoxal, and IEPOX using detailed reactions and reversible pathways in models or observed as reversible processes in chamber experiments, leading to a relatively lower SOA yield compared to the surface-controlled irreversible uptake (Lim et al., 2013; Knote et al., 2014; Galloway et al., 2009; El-Sayed et al., 2018; Budisulistiorini et al., 2017). The non-volatile assumption used in this paper allows an upper-limit estimation of the importance of these additional SOA formation pathways."**

**Comment 21:** *Line 164: change to "was mostly"*
**Response 21:** This sentence has been moved to L162 in the revised manuscript and now it reads as follows:
**"The SOA module mostly follows Pankow et al. (2015)."**

**Comment 22:** *Line 165: Please elaborate more on the non-ideality calculation of the organic-water mixture for non-experts.*
**Response 22:** This sentence has been moved to L199-200 in the revised manuscript and now it reads as follows:
**"POA is also involved in the calculation of activity coefficients for the**

**organic-water mixture.***"*

**Comment 23:** *Line 170: Is this the absorbing organic phase?*
**Response 23:** Yes. The text has been revised to "the absorbing organic phase".

**Comment 24:** *Line 196: Change to "as water condenses".*
**Response 24:** The text has been revised as instructed.

**Comment 25:** *Line 200: Please elaborate more on the "Kelvin effect neglected" for non-experts.*
**Response 25:** The text has been revised in L210-211 as following:
**"Based on the κ-Köhler theory with linearly additive hygroscopic behavior of each component of the mixed particle"**

**Comment 26:** *Line 204: Change to "can be estimated".*
**Response 26:** The text has been revised as instructed.

**Comment 27:** *Line 242: Observations in 8 sub-regions of the domain during which period?*
**Response 27:** The text has been revised in L278-281 as follows:
**"Table 1 lists model statistics of mean observation (OBS), mean prediction (PRE), mean bias (MB), gross error (GE) and correlation coefficient (R) based on WRF and observations at monitoring sites located in 8 sub-regions of the domain (Figure S1) during January and July of 2013."**

**Comment 28:** *Line 251: No significant improvements observed when applying the above additions means that the model is still missing a significant pathway to SOA formation, especially since OA in both seasons are dominated by SOA based on observations. This should be discussed in detail and in the context of previous studies and findings from AMS measurements in China.*
**Response 28:** We have revised the text and added a discussion about the underestimation of SOA in the current model. Please refer to Response 2 for more details.

**Comment 29:** *Line 258: Does it capture the observed diurnal variation? What is the R2 or R of the timeseries of the modeled to observed values? What is the ratio of the two? In many cases, it seems that the difference is higher than a factor of 4. Is that a usual discrepancy? If so, how is much is it improved when incorporating the detailed SOA models?*
**Response 29:** Our model can capture the diurnal variation. The mismatching

of several peak values might be due to uncertainties in the emission inventory and the underestimate of SOA in the current model. The R of the modeled to observed OA is 0.55. The ratio of the averaged prediction to observation is 0.75. Since the model predicts a very small ratio of SOA to POA, the improvement from the detailed SOA model is insignificant. The small SOA/POA ratio might be due to the missing SOA from other pathways including POA aging and oxidation from IVOCs and VOC oxidation products. This has been explained in Response 2.

We revised the text in L301-304 to expand more discussion of the modeled and observed OA comparison:
"**CMAQ can well capture the observed diurnal variation of OA in Beijing during wintertime, except for the underestimates of peak values. The correlation coefficient of modeled to observed OA is 0.55. We find a 25% underestimate of OA on average.**"

**Comment 30:** *Line 259: It would be great if "better" was described with statistical terminology. A way to describe the data and the comparison to modeled values would be to generate box and whiskers of the ratio of observations to modeled values for non-polluted and polluted days, respectively.*
**Response 30:** We did a mistake in the MFB and MFE calculation for OA, which should be -0.28 and 0.54, respectively. We also calculated the biases on polluted and non-polluted days of OA. MFB and MFE of polluted days are -0.38 and 0.64, which are -0.26 and 0.52 for non-polluted days. The text has been revised in L306-309 as follows:
"**The mean fractional bias (MFB) and mean fractional error (MFE) of polluted days are -0.38 and 0.64, respectively, which are worse than that of the non-polluted days (-0.26 for MFB and 0.52 for MFE). The overall MFB and MFE of OA during January are -0.28 and 0.54, within the criteria (MFB≤±0.6; MFE≤0.75) suggested by EPA (2007).**"

**Comment 31:** *Line 262-263: POA is not the primary contributor to OA in Beijing in winter. Many studies show that SOA is the major contributor and the path towards SOA formation is currently unknown and strongly dependent on LWC in the particles. Aging of POA not treated in the model is not guaranteed to be the main source of SOA.*
**Response 31**: Please refer to Response 2.

**Comment 32:** *Line 280: Here only one season is provided in terms of timeseries comparison of the model and obs. Please provide both seasons.*
**Response 32:** Unfortunately, detailed chemical composition measurements for

aerosols are very limited in China during 2013. We only have observations of OC and OA in January of 2013 and PM$_{2.5}$ in July of 2013 available for model evaluation.

**Comment 33:** *Line 285: Figure S5 shows the anthropogenic SOA and not the dicarbonyl SOA. Please separate the contributions and discuss them in the main text. Identifying the contribution of different compounds to SOA formation in China would be of great interest to the scientific community.*
**Response 33:** The contribution of each precursor to SOA of this episode has been shown in Hu et al. (2017) and will not be discussed in detail in the current study. The text has been revised in L344-346 as following:
"**Anthropogenic emissions are the major sources of SOA (Figure S6), such as dicarbonyl products from the oxidation of xylene and toluene (Hu et al., 2017).**

**Comment 34:** *Line 312: What about particulate nitrate?*
**Response 34:** The interactions between water-inorganics and water-organics are treated separately in the current model. We only focus on the water-organic interaction in the current study.

**Comment 35:** *Line 380: RH is higher than 70% all the time. What is the meaning of this sentence?*
**Response 35:** We have removed this figure to avoid confusion.

[revised manuscript text omitted]

---

## Author Comment (AC2) · 16 Apr 2020

**Response to Referee #2**

The authors thank the reviewer for the comments that improve the quality of the paper. The detailed responses are given as follows. The reviewer comments are shown in italic fonts, the responses are in regular font, and the revised text is in bold font.

*In this paper, Li et al. have modified the CMAQ model to take into account the impacts of water partitioning and polarity of organic compounds on SOA formation. The model was applied over Eastern China to estimate the regional and seasonal impacts of these modifications on SOA and the aerosol water content. This study may have potential to contribute in the organic aerosol modeling field but major revisions needs to be done before publication. In particular, I have several concerns regarding the validity of the scientific methodology used and the presentation of the study. Therefore, I would recommend publication only if these comments will be addressed and fundamental changes will be contacted.*

**Major comments**
**Comment 1:** *Page 4 line 94: The majority of current CTMs have replaced the 2-product model with the VBS approach. Please make this clear and refer to the 2-product model of Odum et al. (1996) for historical reasons.*
**Response 1:** The text has been revised in the manuscript (L88-95) to make this clear:
"**The formation of condensed organic products is commonly represented by lumped surrogate SVOCs in a 2-product model with volatilities and SVOC yields fitted to chamber experiments (Odum et al., 1996). To better represent the volatility of primary organic aerosol (POA) and the multi-generation oxidation of SVOCs to a wider range, Donahue et al. (2006) proposed the volatility basis set (VBS) model in which the mass yields of SVOCs are fitted to a fixed number of volatility bins (usually 0.01-105 μg m-3). The VBS model has been adopted by several CTMs (such as WRF-Chem, GEOS-Chem, etc.).**"

**Comment 2:** *Page 6, line 162: Please add the appropriate references to support the nonvolatile nature of the products by these oxidation pathways.*
**Response 2:** Formation of these non-volatile SOA was traditionally treated in the CMAQ model, except for dicarbonyls, IEPOX, and MAE that are assumed to form SOA by irreversible reactive uptake in our model, as an upper-limit estimation of SOA from these precursors. The text has been revised to make this clear (L162-173 in the manuscript):
"**The SOA module mostly follows Pankow et al. (2015). Two types of SOA as traditionally treated in CMAQ were considered, "semi-volatile" (SV) portion that formed via equilibrium absorption-partitioning of SVOCs, and**"

"non-volatile" (NV) portion that includes the oligomers and SOA formed via direct oxidation of aromatics at low-NOx. SOA from dicarbonyls, IEPOX, and MAE were formed by irreversible reactive uptake and categorized as NV-SOA in the current model as well. Some studies investigated SOA from glyoxal, methylglyoxal, and IEPOX using detailed reactions and reversible pathways in models or observed as reversible processes in chamber experiments, leading to a relatively lower SOA yield compared to the surface-controlled irreversible uptake (Lim et al., 2013; Knote et al., 2014; Galloway et al., 2009; El-Sayed et al., 2018; Budisulistiorini et al., 2017). The non-volatile assumption used in this paper allows an upper-limit estimation of the importance of these additional SOA formation pathways."

**Comment 3:** *Page 6 line 149: There is no discussion in the methodology about the observations and the statistical analysis metrics used to evaluate the model performance. Especially for the OA observations, there is no reference provided or description of the methods used.*

**Response 3:** Thank you for pointing this out. We have included details of observation data, statistical analysis metrics for both meteorology and aerosols in the revised manuscript:

In L265-276

"**The meteorological inputs and emissions have been used in several previous publications. Model performance on meteorological parameters (temperature and RH), gaseous species and gas and aerosol concentrations have been extensively evaluated (Hu et al., 2016; Hu et al., 2017; Qiao et al., 2018; Shi et al., 2017). A summary of the model performance related to this study is provided below. Observed meteorological data were obtained from the National Climatic Data Center (ftp://ftp.ncdc.noaa.gov/pub/data/noaa). Observations of OC at two urban locations, Beijing (Cao et al., 2014; Wang et al., 2015) and Guangzhou (Lai et al., 2016) and OA in Beijing (Sun et al., 2014) during January of 2013 as well as surface PM$_{2.5}$ at several monitoring sites during July of 2013 from China National Environmental Monitoring Center (http://113.108.142.147:20035/emcpublish/) were used to evaluate model estimates of aerosols. Details of measurement methodology and uncertainties of observations are listed in the corresponding references.**"

In L278-282

"**Table 1 lists model statistics of mean observation (OBS), mean prediction (PRE), mean bias (MB), gross error (GE) and correlation coefficient (R) based on WRF and observations at monitoring sites located in 8 sub-regions of the domain (Figure S1) during January and July of 2013. The benchmarks for the MM5 model (another meteorology model) of 4-12km horizontal resolution suggested by Emery et al. (2001) are also listed in**

the table."

In L294-296
"**Overall, the mean fractional bias (MFB) and mean fractional error (MFE) of OC are -0.20 and 0.27, within the criteria (MFB≤±0.6; MFE≤0.75) suggested by EPA (2007).**"

In L308-309:
"**The overall MFB and MFE of OA during January are -0.28 and 0.54, within the criteria (MFB≤±0.6; MFE≤0.75) suggested by EPA (2007).**"

**Comment 4:** *Page 7 line 179: More details are needed here. (1) How the model defines the low NOx and high NOx conditions? Which compounds each of the lump species represent? What is the difference between the lumped species of the same precursor (e.g., BNZ1, BNZ2, BNZ3)? (2) Can you include the aerosol yields for each lumped species in tables S1 and S2? Are these aerosol yields NOx-dependent? (3) What does the SVP stands for in Tables S1 and S2?*

**Response 4:** (1) SOA formation in CMAQv5.0.2 is based on the frame of a previous version 4.7.1. All the details about "high" and "low" $NO_x$ conditions (based on chamber experiments of corresponding VOCs), lumping species and method of each precursor, and the yields of precursors from parent VOCs have been documented by Carlton et al. (2010) and summarized in the revised supplemental materials as following:

"**The CMAQ model treats high and low NOx SOA formation pathways during OH oxidation by allowing the lumped RO2 radical to competitively react with HO2 and NO. Using the lumped ARO1 species as an example, an SOA formation specific RO2 radical ARO1RO2 is added as a gas phase reaction product with OH:**

$$ARO1 + OH \rightarrow ARO1RO2 + products$$

**The ARO1RO2 can react with both HO2 and NO, as shown in the following two reactions:**

$$ARO1RO2 + HO2 \rightarrow HO2 + TOLNRXN; \quad k1$$
$$ARO1RO2 + NO \rightarrow NO + TOLHRXN; \quad k2$$

**Details of the determination of the rate constants can be found in Carlton et a. (2010). The TOLNRXN and TOLHRXN are counter species that track how much ARO1 is reacted through low NOx and high NOx pathways, respectively, in one gas chemistry time step. The concentrations of these counter species are passed into the aerosol module to calculate the formation semi-volatile products (TOL1 and TOL2) in the high NOx pathway and non-volatile products (TOL3) in the low NOx pathway, using the mass-specific yields, as listed in Table S1 and S2. Equilibrium partitioning of TOL1 and TOL2 in the gas phase and their counterparts ATOL1 and ATOL2 in the organic phase are affected by temperature and the amount of absorbing organics in the aerosol phase. Similar**

**treatments are applied to the other lumped aromatic compounds ARO2, with xylene as a representative and most abundant species in that group, and to benzene. SOA formation from lumped long-chain alkene species ALK5, and isoprene and monoterpenes are not considered as NO$_x$ dependent and are represented by equilibrium partitioning of one or two semi-volatile oxidation products. Details of the mass-specific yields of semi-volatile products and other related parameters can be found in Table S1 and S2.**"

We revised the text in (L188-190) to make it clear:
"**More details about the lumped precursors such as formation conditions ("high" and "low" NOx), lumping species and method, and yields from parent VOCs can be found in Carlton et al. (2010) and summarized in SI.**"

(2) In the CMAQ model, the amount of SOA can form after a precursor reacts with OH, O$_3$ or NO$_3$ depends on the volatility of the products, which is temperature dependent, and the amount of the absorbing organics. The mass yields of the semi-volatile or non-volatile products are included in Table S1 and S2 in the revised supplementary materials. For more details, we refer the readers to Carlton et al. (2010) and the references therein.

(3) SVP is the saturation vapor pressure. We have explained this in the corresponding tables.

**Comment 5:** *Page 8, lines 194-198: (1) Does the absorbing phase of equation 1 includes only the water associated with the organics (from eq. 3) or it includes the total water (including the water associated with the inorganic aerosol components)? (2) Under high RH (higher than the organic/inorganic phase separation RH, SRH), the aerosol organic phase is well mixed with the inorganic salts and, therefore, the aerosol water associated with the inorganic constituents can also contribute to the SOA absorbing medium (Pye et al., 2017). Please clarify what you have assumed here and add the relative discussion.*
**Response 5:** (1) The absorbing phase of equation 1 only includes organic and water associated with organics when considering water-organic interactions. We explained this in the original text L195-196:
"In addition to organic compounds, water partitioning into OPM is enabled according to Eq 1 and Eq 2. In such a case, the absorbing phase in Eq 1 includes both organic aerosols and water partitioning into OPM."

(2) We didn't consider the mixing of organic and inorganic phase in this study and assumed that they are always two distinct aerosol phases without direct interactions. The phase separation RH (SRH) depends on the OM/OC ratio of the organic phase. Unlike the conditions modeled by Pye et al. (2017) for the

southeast US where SOA is often the dominant OA component, the winter episode we modeled is dominated by primary emitted organic aerosols thus with a relatively low OM/OC (~1.4-1.6). The SRH based on equation (7) of Pye et al. (2017) is ~97%-99%. The summer episode has more contributions of SOA to OA, with OM/OC~1.8, which corresponds to an SRH of 87%. Thus, we don't expect interactions of organic and inorganic phases to occur in high frequency to greatly influence the model results. We assumed no interactions between inorganic and organic phases in the current model. We have also revised text in L207-208:

"**In the current model, we assumed no interactions between the inorganic and organic phases.**"

**Comment 6:** *Page 8, Equation 3: This equation gives the volume of water associated with the organic fraction of the aerosol. However, ALW on the left-hand side refers to the mass of water. Please correct.*

**Response 6:** The equation has been corrected in the revised text L212-214:
"

$$ALW_{org} = \rho_w V_{org} \kappa_{org} \frac{a_w}{1 - a_w} \qquad \textbf{(Eq3)}$$

**where $\rho_w$ is the density of water (assumed to be 1 g cm$^{-1}$), $V_{org}$ is the volume concentration of organics, and $a_w$ is the water activity (assumed to be the same as RH).**"

**Comment 7:** *Page 8, lines 203-204: How do you estimate the hygroscopicity?*

**Response 7:** We used Eq3 to estimate the hygroscopicity. This has been clarified in the revised text in L214-216:

"**Since $ALW_{org}$ in this study is calculated mechanistically using the partitioning theory, $\kappa_{org}$ can be estimated by rearranging Eq3:**

$$\kappa_{org} = \frac{ALW_{org}}{\rho_w V_{org}} \times \frac{1 - a_w}{a_w} \qquad \textbf{(Eq4)}$$

"

**Comment 8:** *Page 8, lines 204-205: How do you calculate the ALW? Are you using $\kappa_{org}$ and the eq3? Do you use the kappa hygroscopicity to calculate the ALW? If so, how you estimate the kappa?*

**Response 8:** In this study, the $ALW_{org}$ is independently calculated by mechanistically allowing water molecules to partition into the organic phase with UNIFAC calculated activity. In such a case, Eq 3 can be used to provide an independent estimation of $\kappa_{org}$. Linear regression analysis can be performed using the calculated $\kappa_{org}$ against the model calculated O/C ratios, as shown in Figure 4. We have removed this sentence to avoid confusion.

**Comment 9:** *Page 9, lines 212-216: Add a reference to tables S1 and S2. Furthermore, why the values of OM:OC in tables S1/S2 are different than the values provided by your reference (Pye et al., 2017)?*

**Response 9:** We used OM:OC ratio in Pankow et al. (2015). There were mistakes in the original Table S1 and S2 and have been corrected in the revised supplemental materials. The text has also been revised in L228-229 as following:

"**The OM:OC ratio of each SOA component follows Pankow et al. (2015) as shown in Table S1-S2.**"

**Comment 10:** *Page 9 lines 217-218: Can you add the total size of the model domain?*

**Response 10:** The text has been revised in L231-233 as follows:

"**The simulation domain has a horizontal resolution 36 km × 36 km (100 × 100 grids) and a vertical structure of 18 layers up to 21 km, which covers eastern China as shown in Figure S1.**"

**Comment 11:** *Page 9 lines 233: (1) What boundary conditions are used? Please make a comment on how these can affect the simulation results. (2) I would recommend adding spatial maps of primary organic aerosol emissions and SOA precursor emissions and summarizing in a Table the domain average emission rates of POA and each SOA precursors.*

**Response 11**: (1) We used a predefined boundary profile in CMAQ that represents a clean continental condition. The northern and western boundaries, as well as areas to the further north and west, are mostly remote areas with much lower emissions. The mountains in the north and west part of the domain also limit the influence of emissions enter from the boundaries to the central part of the domain. The influence of marine air from the south and east boundaries is also small, as local emissions dominate the concentrations.

(2) A figure and a table showing emissions of POA and SOA precursors were added to the revised supplemental materials as Figure S2 and Table S4.

**Comment 12:** *Pages 9-10 lines 234-238: (1) This paragraph needs to be expanded and written in a separate section. In this section the authors should describe in more detail the following: i) Basecase simulation. Please explain how the default CMAQ is simulating POA and SOA and how different is this modelling configuration with the one the authors are testing, ii) Sensitivity simulations. Please explain in much more detail the sensitivity simulations conducted in this work. (2) In addition the authors say that they have conducted three sensitivity scenarios named S1, S2 and S3. In their manuscript they only show results from S3 and they never discuss the results of S1 and S2.*

**Response 12:** (1) Details of how default CMAQ simulates SOA formation through equilibrium partitioning of lumped semi-volatile products into the

organic phase (which include both POA and SOA) have been described by Carlton et al. (2010) and Hu et al. (2017) so we don't think it is necessary to repeat it here. We expanded each simulation scenario in details in the revised text in L249-259:

"**Four scenarios are investigated in this study. The base case (BS) applies the default secondary organic aerosol module of CMAQ v5.0.1. In this case, no water partitioning into OPM is considered. Lumped semi-volatile products from the oxidation of various precursors partition into a single organic phase, which is considered as an ideal mixture of POA and SOA with $\gamma_{org}$=1. The water case (S1) includes water partitioning into OPM, which is again considered as an ideal solution ($\gamma_{org}$=1 and $\gamma_{H2O}$=1). The UNIFAC case (S2) considers the interaction between organic constituents with UNIFAC calculated activity coefficients ($\gamma_{org}\neq$1) but does not allow water partitioning into OPM. The combined case (S3) allows both water partitioning and interactions between all constituents (including water and organics) using UNIFAC calculated activity coefficients ($\gamma_{org}\neq$1 and $\gamma_{H2O}\neq$1).** "

(2) The impacts of S1 and S2 were discussed in Section 4 of the original manuscript Page 16 L405-419. We also included a description of the outlines of the results and discussions in the original manuscript Page 6 L140-148. The text has been revised in L259-262 to make it clear:

"**The results of BS and S3 are used to examine the overall impacts of water partitioning into OPM and polarity of organics on SOA and ALWorg, as shown in Section 3.1-3.4. The separate influences of those two processes on SOA from S1 and S2 are discussed in Section 3.5.**"

**Comment 13:** *Page 10 line 240: The section "Model evaluation" is extremely problematic and raises questions on the validity of the modelling results given that the model evaluation is insufficient. More specifically: 1) Given that the CMAQ default configuration has been modified to consider the importance of water and organic compound polarity on SOA formation, an accurate evaluation of the updated model performance is needed. 2) You should compare the model results for organic compounds during both July and January 2013. Currently, the evaluation includes a comparison with OC observations only during January over only three locations of the relatively large model domain. Furthermore, please mention in the text what factor have you used to convert the modeled OA to OC. 3) The total PM2.5 measurements have been used to evaluate the model performance during July without explaining the rationale of this choice since the focus of this study is solely the organic fraction of the aerosols. I suggest removing the PM2.5 evaluation or at least moving it to the supplement. 4) Can you include more OC/OA observations over other areas of their domain in your evaluation? 5) It is also important to compare the simulated*

*POA and SOA against observations (e.g., from AMS). Furthermore, it would be helpful to show how the model performance against SOA measurement changes between the BC and the S1, S2, S3 cases.*

**Response 13:** (1) For the updates in CMAQ except for water partition into OPM and non-ideality of the organic-water mixture, previous studies have extensively examined the model performances and will not be further discussed in detail in this work. The text has been revised in L149-161 to reflect this:

"**Model configurations were largely based on that used by Hu et al. (2016) as summarized below. Firstly, SAPRC-11 was expanded for a more detailed treatment of isoprene oxidation and tracking dicarbonyl products (glyoxal and methylglyoxal) from different groups of major precursors (Ying et al., 2015). Secondly, SOA from isoprene epoxydiols (IEPOX), methacrylic acid epoxide (MAE) and dicarbonyls through surface-controlled irreversible reactive uptake were added (Hu et al., 2017; Li et al., 2015; Liu et al., 2020; Ying et al., 2015). Thirdly, the heterogeneous formation of secondary nitrate and sulfate from NO$_2$ and SO$_2$ reaction on the particle surfaces (Ying et al., 2014) were added, which is an important source of secondary inorganic aerosols (Zheng et al., 2015) and improves model estimates of nitrate and sulfate (Qiao et al., 2018; Shi et al., 2017). Fourthly, SOA yields were corrected for vapor wall loss (Zhang et al., 2014). Impacts of the above updates on model performances have been extensively discussed in the cited work and will not be further investigated in the current study.**"

(2) Unfortunately, detailed chemical composition measurements for aerosols are very limited in China during 2013. We only have observations of OC and OA in January of 2013 and PM$_{2.5}$ in July of 2013 available for model evaluation. Thus, we opt to use the most relevant data to provide a very limited assessment of the capability of the model in predicting SOA. The factors for OA to OC conversion follow the OM:OC ratio listed in Table S1 and S2 for SOA. POC is directly predicted by the model. The text has been revised in L291-292 to make this clear:

"**The factors used to convert SOA to OC (SOC) are listed in Table S1-S2. OC from POA (POC) is directly predicted by the model.**"

(3) Even though limited OC/OA measurements are available to us during this period, the base case model is later applied by another research group to model wintertime SOA formation in east China (Liu et al., 2020). The predicted OC and SOA agree well with observations (Figure 2 of Liu et al. 2020), and the model performance statistics for OC and SOA are similar to those of PM$_{2.5}$. We agree with the reviewer that PM$_{2.5}$ is not an ideal indicator to evaluate the capability of the model in predicting SOA, however, as a significant fraction of PM$_{2.5}$ in July is secondary, this still provides an indirect assessment of the model prediction of oxidation capacity of the atmosphere, which is import for

SOA formation. Additional modeling studies are needed to evaluate the performance of the model in summer.

We have explained this in the revised text (L327-329) as following:

"**Due to the lack of observed OC and OA in July of 2013, as an alternative, model performances are evaluated by comparing predicted and observed PM$_{2.5}$ at ground sites (Figure S1) as shown in Figure S3.**"

(4) We do not have more OC and OA data for the simulating episode of 2013.

(5) We did not have SOA observations in this episode. We compared the modeled SOA/POA ratio with AMS observations from other literature, finding a significant underestimation in the current model. This bias might be due to missing SOA from combustion (intermediate volatile organic compounds, IVOCs) and not treating POA as semi-volatile. We added a discussion about the model bias in the revised manuscript in L309-326:

"**Again, no apparent changes of SOA nor OA are observed between case S3 and BS (not shown), since POA is predicted to be the primary contributor to OA at Beijing in winter in the current model, with an averaged SOA/POA ratio of 0.12. This ratio is much lower than the field observation of about 0.45-1.94 (Zhao et al., 2019; Sun et al., 2013; Sun et al., 2016). The bias might be due to the missing SOA converted by partitioning and aging of semi-volatile POA as well as oxidation from intermediate volatile organic compounds (IVOCs) and VOC oxidation products. Those pathways are shown to be important for SOA formation by modeling, field and chamber studies (Hodzic et al., 2010; Jimenez et al., 2009; Murphy et al., 2017; Robinson et al., 2007; Shrivastava et al., 2008; Tkacik et al., 2012; Zhao et al., 2014; Zhao et al., 2016a).**

**A sensitivity test was performed by using the newest CMAQ model version 5.3.1 that includes all the above processes in the aerosol module. The SOA/POA ratio in Beijing is improved greatly to be 1.53 in winter. However, high uncertainties still exist in the emissions of the involved precursors and characterization of SOA formation through these processes, needing further constrains by observations. Their influences on water partitioning into OPM and non-ideality of the organic-water mixture on SOA will be evaluated in a future study.**"

Since SOA is underestimated, no significant differences in BS and S3 are observed. Case S1 and S2 are designed for the sensitivity test of model results to water partitioning and non-ideality of condensed organics separately. Therefore, we did not evaluate model performances from S1 and S2.

**Comment 14:** *Page 10 lines 251-253: The authors state here that the impacts of water-cocondensation and polarity of organic condensed species on SOA*

*formation are not significant during winter. This highlights the need to evaluate their model results during July where they have found significant changes with the basecase simulation. Furthermore, the results from the three sensitivity simulations should be evaluated individually.*

**Response 14**: Unfortunately, we have no observations of OC, OA or (and) SOA of July 2013. We tried to evaluate model performances by comparing the predicted and observed PM$_{2.5}$ as an alternative. This has been explained in the revised manuscript in L327-329:

"**Due to the lack of observed OC and OA in July of 2013, as an alternative, model performances are evaluated by comparing predicted and observed PM2.5 at ground sites (Figure S1) as shown in Figure S3.**"

Since SOA is underestimated, no significant differences in BS and S3 are observed. Case S1 and S2 are designed for the sensitivity test of model results to water partitioning and non-ideality of condensed organics separately. Therefore, we did not evaluate model performances from S1 and S2.

**Comment 15:** *Page 10 lines 254-256: These are indeed possible factors. Can the authors comment, based on their analysis, which of these two possible factors is more important and try to be more specific? A comparison with AMS observations would be helpful here.*

**Response 15:** We did a sensitivity test by simulating the same episode with the current CMAQv5.3.1 in which POA was treated as semi-volatile and aging in the gas phase. Also, a missing source of SOA from intermediate VOCs (IVOCs) oxidation and aging of IVOCs and VOCs oxidation products (pcSOA) was added in CMAQv5.3.1. The results showed that the modeled SOA/POA has been improved from 0.12 of case S3 to 1.53 (Figure R1), more close to AMS observations. However, there were still some peak values underestimated by the model, which might be due to the uncertainties of POA emissions. We have revised the text (L309-326), as mentioned in Response 13 (5).

[Figure]

**Figure R1.** Modeled concentration of semi-volatile POA (sv-POA), OA and fraction of each organic aerosol component fsvPOA (sv-POA), foxPOA (oxidation of sv-POA), fpcSOA (pc-SOA) and fsvSOA (traditional SOA) in Beijing (a, c) and Guangzhou (b, d). Observations of OA in January 2013 at Beijing (Obs.) are also included in (a). The left axis is the concentration ($\mu g$ m$^{-3}$) and the right axis is the fraction of OA components.

**Comment 16:** *Page 11 line 263: The aging of POA, under specific conditions can enhance the SOA formation, especially over polluted areas. Can the authors comment how this important omission of their model configuration can affect their result? Once again, a comparison against POA and SOA from AMS observations will be helpful to identify the limitations of their model due to the treatment of POA as non-volatile and non-reactive.*

**Response 16:** We did a sensitivity test by CMAQv5.3.1 that includes SOA from POA aging, IVOCs oxidation, and aging of IVOCs and VOCs oxidation products. The modeled SOA/POA is improved greatly in Beijing with no significant improvement in OA compared to the results of case S3. A significant improvement of SOA was observed from the contribution from IVOCs oxidation, and the aging of IVOCs and VOCs oxidation products (pc-SOA). Discussions about this have been included in the revised text in L309-326, as mentioned in Response 15.

Also, we examined the sensitivity of SOA and organic liquid water (ALW$_{org}$) to pcSOA and POA in an offline calculation in Beijing, Guangzhou, Jinan, and Nanjing. POA has the same properties as we used in the model. Non-volatile isoprene SOA is taken to represent pcSOA as their similarities in saturation vapor pressure and O:C ratio. We found that both SOA and ALW$_{org}$ are

positively correlated with pcSOA, increased by 2-5 times in different locations when pcSOA increased by 2 times.

**Comment 17:** *Page 11 line 264: Please provide two spatial maps of the fraction SOA/Total OA during January and July 2013 so as to show the contribution of POA and SOA to total OA during each simulation period.*

**Response 17:** We have added a figure of SOA/OA ratio by case BS and S3 (as shown below) in the revised supplemental materials (Figure S13).

[Figure]

**Figure R2**. Averaged SOA/OA ratio from case BS and S3 during January and July of 2013.

**Comment 18:** *Page 13 line 317: You need a zonal map to show how the water partition changes with altitude and not the total column.*

**Response 18:** Most of the SOA and ALW$_{org}$ retain in the lower levels of the troposphere. The information on altitude variation may not be very useful. Thus, no changes were made regarding this comment. We have also deleted this sentence in the revised manuscript to avoid confusion.

**Comment 19:** *Page 13, lines 322-336: It is not clear how you calculate the $\kappa_{org}$ in your model. This is very important for this section.*

**Response 19:** We explained this in the revised text in L214-216:

"**Since ALW$_{org}$ in this study is calculated mechanistically using the partitioning theory, $\kappa_{org}$ can be estimated by rearranging Eq3:**

$$\kappa_{org} = \frac{ALW_{org}}{\rho_w V_{org}} \times \frac{1 - a_w}{a_w} \qquad \textbf{(Eq4)}$$

"

**Comment 20:** *Figure 1: The quality of the figure is poor. It is extremely difficult to see all the plotting data and the changes due to the use of different scenarios (especially in figure 1b).*

**Response 20:** We removed the results of BS since they are very similar to those of S3. Now the figure has been revised as follows:

[Figure]

**Figure 1.** Comparison of (a) observed and modeled organic carbon concentration at University of Beihang (BH), Tsinghua University (TH) and Guangzhou (GZ); (b) observed organic aerosol (Obs-OA) at Beijing and predictions of total OA (pOA) and SOA (pSOA), unit is µg m-3. Locations of monitoring sites are shown in Figure S1.

**Comment 21:** *Figure 2: I found the use of daily maximum concentration in the*

*"difference" maps misleading. Given that you have the monthly average SOA from the basecase simulation, I would prefer to see the absolute (and relative) change of the monthly average SOA due to the use of S3 as well, and not the daily maximum. Furthermore, please add (a), (b), etc. to each subplot of the figure and add this information to the figure caption (apply this change in the rest of the figures as well).*

**Response 21:** The monthly-averaged daily maximum differences have been replaced by the monthly-averaged differences for all the corresponding figures to reflect the general impacts on SOA and ALW$_{org}$. Each panel of the figure is labeled in sequence.

**Comment 22:** *Figure 4: All the fitted correlations listed here suggest that compounds with very low or zero O:C have negative hygroscopicity. Can you comment on this limitation and include a discussion in the text?*

**Response 22:** The relatively low values of hygroscopicity for low O:C ratio might be due to the linear regression. We also did an exponential fitting for the two variables so that the hygroscopicity falls in the range of (0,1) and is positively correlated with O:C ratio. The text and figure 4 have been revised accordingly.

In L402-407:

"**In both seasons, $\kappa_{org}$ approaches zero and negative values as O:C decreases, which might be due to the linear regression of $\kappa_{org}$ and O:C. To avoid this, an exponential fitting of the two variables is performed so that $\kappa_{org}$ falls in the range of (0,1) and is positively correlated with O:C. In this case, the fitted correlations are $\kappa_{org}=1-exp(-(O:C/1.88)^{2.29})$ and $\kappa_{org}=1-exp(-(O:C/1.06)^{4.50})$ for January and July of 2013, respectively.**"

[Figure]

**Figure 4**. The correlation of hygroscopicity of organic aerosol ($\kappa_{org}$) and O:C ratio at 9 representative cities including Shenyang (SS), Beijing (BJ), Jinan (JN), Zhengzhou (ZZ), Xi'an (XA), Nanjing (NJ), Shanghai (SH), Chengdu (CD), and

Guangzhou (GZ) in January (a) and July (b) of 2013. O:C ratios are categorized into 10 bins. In each bin, the ranges of O:C and $\kappa_{org}$ are represented by bars. The mean values of O:C and $\kappa_{org}$ are represented by triangles colored by the averaged RH of each bin. The relationship between $\kappa_{org}$ and O:C is fitted by a linear function with reduced major axis regression (blue lines) and an exponential function (red lines), respectively. $\kappa_{01}$ and $\kappa_{07}$ represent the fitted correlation for January and July, respectively.

**Comment 23:** *Table S3. Please explain in the table what the fraction in the last column stands for. How have you estimated the molecular weight and fraction of the POA from unknown compounds?*
**Response 23:** The last column is the molar fraction of each POA surrogate. We have clarified this in the first row of this column in the table. The molecular weight and molar fraction of unknown compounds of POA have been listed in the original table already, which are 390 and 0.3, respectively.

**Minor Comments**
**Comment 24:** *The language and structure of the sentences can be substantially improved in many parts of the manuscript. Just a few examples are listed here, but I suggest revising thoroughly the wording in the whole text.*
**Response 24:** We thank the reviewer for pointing this out. The whole text, as well as figures and tables, have been revised carefully.

**Comment 25:** *Page 5, lines109-11: Please rephrase. The sentence sounds wrong.*
**Response 25:** The text has been revised in L109-111 as follows:
"**Laboratory and field studies have observed water absorbed by SOA from a variety of precursor VOCs (Lambe et al., 2011; Zhao et al., 2016b; Asa-Awuku et al., 2010; Varutbangkul et al., 2006).**"

**Comment 26:** *Page 5, lines117-118: Please rephrase.*
**Response 26:** The sentence has been revised in L116-117 as follows:
"**The total water content is the summation of water associated with each solute at the same water activity.**"

**Comment 27:** *Page 6 line 144: OC and OA abbreviations have not been used before in the main text.*
**Response 27:** OA abbreviation has been explained in a previous part of the revised manuscript in L120-122:
"**Pye et al. (2017) found that the modeled organic aerosol (OA) improved significantly but biased high at nighttime when ALW$_{org}$ is included in the calculation.**"

Thus, this sentence has been revised in L139-141 as follows:

"**The model performance was evaluated against observed meteorological parameters (temperature and relative humidity, RH) as well as PM$_{2.5}$, organic carbon (OC), and OA at ground monitoring sites.**"

**Comment 28:** *Page 8, line 204: Please correct the "can estimated" to "can be estimated".*
**Response 28:** The text has been revised as instructed.

**Comment 29:** *Page 8 line 206: I would use the word "correlate" instead of "dependent"*
**Response 29:** The text has been revised in L219-229 as follows:

"**In many studies, $\kappa_{org}$ is assumed to increase linearly with the oxidation state of OA, expressed as the O:C ratio (Massoli et al., 2010;Duplissy et al., 2011;Lambe et al., 2011).**"

**Comment 30:** *Page 9 line 229: Change "in" with "on"*
**Response 30:** The text has been revised as instructed.

**Comment 31:** *Page 10 line 254: Change the sentence to: "In Beijing and Guangzhou, these impacts are not significant during winter"*
**Response 31:** The sentence has been revised in L298-300 as:
"**No significant differences in OC are observed in S3 compared to BS (not shown), likely due to the biased-low SOA predicted in the current model so that limits the impact of ALW$_{org}$ on SOA formation.**"

**Comment 32:** *Page 11 line 274-275: Please rephrase.*
**Response 32:** The sentence has been revised as follows:
"**The criteria of MFB and MFE followed recommendations by Boylan and Russell (2006).**"

**Comment 33:** *Page 11 line 282: Which two areas? You have mentioned several areas in the previous sentence.*
**Response 33:** The sentence has been revised as follows:
"**Monthly-averaged SOA concentrations in the above areas are up to 25 and 15-20 µg m$^{-3}$, respectively.**"

**Comment 34:** *Page 13 line 317: "column water". Please rephrase*
**Response 34:** The text has been revised in L378 as follows:
"**Based on the column concentrations of ALW$_{org}$ and ALW$_{org}$/SOA ratio (Figure S8),**"

**References**

Carlton, A. G., Bhave, P. V., Napelenok, S. L., Edney, E. O., Sarwar, G., Pinder, R. W., Pouliot, G.

A., and Houyoux, M.: Model Representation of Secondary Organic Aerosol in CMAQv4.7, Environ. Sci. Technol.,, 44, 8553-8560, 10.1021/es100636q, 2010.

Hu, J., Wang, P., Ying, Q., Zhang, H., Chen, J., Ge, X., Li, X., Jiang, J., Wang, S., Zhang, J., Zhao, Y., and Zhang, Y.: Modeling biogenic and anthropogenic secondary organic aerosol in China, Atmos. Chem. Phys., 17, 77-92, 10.5194/acp-17-77-2017, 2017.

Liu, J., Shen, J., Cheng, Z., Wang, P., Ying, Q., Zhao, Q., Zhang, Y., Zhao, Y., and Fu, Q.: Source apportionment and regional transport of anthropogenic secondary organic aerosol during winter pollution periods in the Yangtze River Delta, China, Sci. Total Environ., 710, 135620, https://doi.org/10.1016/j.scitotenv.2019.135620, 2020.

Pankow, J. F., Marks, M. C., Barsanti, K. C., Mahmud, A., Asher, W. E., Li, J., Ying, Q., Jathar, S. H., and Kleeman, M. J.: Molecular view modeling of atmospheric organic particulate matter: Incorporating molecular structure and co-condensation of water, Atmos. Environ., 122, 400-408, https://doi.org/10.1016/j.atmosenv.2015.10.001, 2015.

Pye, H. O. T., Murphy, B. N., Xu, L., Ng, N. L., Carlton, A. G., Guo, H., Weber, R., Vasilakos, P., Appel, K. W., Budisulistiorini, S. H., Surratt, J. D., Nenes, A., Hu, W., Jimenez, J. L., Isaacman-VanWertz, G., Misztal, P. K., and Goldstein, A. H.: On the implications of aerosol liquid water and phase separation for organic aerosol mass, Atmos. Chem. Phys., 17, 343-369, 10.5194/acp-17-343-2017, 2017.

---

## Author Comment (AC3) · 16 Apr 2020

There are two mistakes in the previous submitted replies need to be corrected. (1) The SOA/POA ratio from CMAQv5.3.1 in Beijing shuold be 0.83. (2) the SOA/OA ratio in Figure R1 has been updated.

Sorry for the inconvenience.

———————————————

**Fig. 1.** Mean SOA, SOA/OA, and pcSOA/SOA ratio predicted during January and July of 2013 by CMAQv5.3.1.

---

## Author Comment (AC4) · 16 Apr 2020

There is one mistake in the previous submitted replies needs to be corrected. The SOA/POA ratio from CMAQv5.3.1 in Beijing should be 0.83.

Sorry for the inconvenience.